



# A Comprehensive Reappraisal of Long-term Aerosol Characteristics, Trends, and Variability in Asia

Shikuan Jin[1], Yingying Ma[1,2], Zhongwei Huang[3], Jianping Huang[3], Wei Gong[1,2,4], Boming Liu[1], Weiyan Wang[1], Ruonan Fan[1], Hui Li[4]

[1]State Key Laboratory of Information Engineering in Surveying, Mapping and Remote Sensing, Wuhan University, Wuhan 430072, China.
[2]Collaborative Innovation Center for Geospatial Technology, Wuhan 430079, China
[3]Key Laboratory for Semi-Arid Climate Change of the Ministry of Education, College of Atmospheric Sciences, Lanzhou University, Lanzhou 730000, China
[4]School of Electronic Information, Wuhan University, Wuhan 430072, China.

*Correspondence to*: Yingying Ma (yym863@whu.edu.cn)

**Abstract.** Changes of aerosol loadings and properties are of importance to understand atmospheric environment and climate change. This study investigates the characteristics and the long-term trends of aerosols of different sizes and types in Asia from 2000 to 2020, by considering multi-source aerosol data, novel analysis method and perspective, all this groundwork

promote the acquisition of new discoveries that are different from the past. The geometric mean aggregation method is applied, and serial auto-correlation is considered to avoid overestimation of trend significance. Among regions in Asia, high values of aerosol optical depth (AOD) are mainly concentrated in East Asia (EA) and South Asia (SA), closely related to population density. The AOD in EA showed the most significant negative trend with a value of $-5.28 \times 10^{-4}$ per year, mainly owing to decreases in organic carbon (OC), black carbon (BC), and dust aerosols. It is also worth noting that this observed

large-scale decrease in OC and BC is a unique and significant phenomenon to region of EA, and mainly around China. By contrast, the aerosol concentrations in SA generally show a positive trend, with an increase value of AOD of $1.25 \times 10^{-3}$ per year. This increase is mainly due to large emission of fine-mode aerosols, such as OC and sulphate aerosol. Additionally, the high aerosol loading in north SA has lower AOD variability comparing with that of East China plain, revealing a relatively more persistent air pollution situation. Over the whole Asia region, the characteristics of percentage changes in different type

AOD are increases in BC (6.23%) and OC (17.09%) AOD with a decrease in dust (-5.51%), sulphate (-3.07%), and sea salt (-9.80%) AOD. Except for anthropogenic emission, the large increase in the percentage of OC is also owing to wild fires found in Northern Asia in the summer. Whereas, the different size AOD only shows slight changes in Asia, that small-size AOD decreases (-3.34%), and the total AOD did not show a significant change, suggesting that, from a trend perspective, decreases in aerosol in recent years have mostly been offsetting earlier increases in anthropogenic emission over Asia. To

summarize, the above findings analyse the comprehensive characteristics of aerosol distributions and reappraise the long-term trends of different aerosol parameters, which will greatly enhance the understandings of regional and global aerosol environment and climatology, as well as fill in the gaps and break through the limitation of past knowledge.



# 1 Introduction

Aerosols are small solid and liquid particles suspended in the atmosphere, originating both from anthropogenic and natural

activities. As one of the critical components of the atmosphere, the increase in anthropogenic aerosol emissions can pose a threat to the ecological environment and public health (Lelieveld et al. 2015; Tie et al. 2009), further, they also influence the Earth's energy balance, not only by scattering and absorbing extraterrestrial solar radiation, but also by changing cloud microphysical characteristics, such as the number and radius of cloud droplets in the form of cloud condensation nuclei (Eck et al. 2010; Nakajima et al. 2007; Rosenfeld et al. 2019). Because of the complex properties and uneven distributions caused

by physical and chemical processes in the atmosphere, aerosol particles can vary over time. Therefore, it is necessary to understand the aerosol spatial distribution and temporal evolution, thus supply the effective approach and data for comprehending the atmospheric environment and global climate changes (Kaufman et al. 2005; Mallet et al. 2013; Solomon 2007).

Asia is the focus of aerosol study, which is a large continent with more than half of the world's population (**Figure 1**).

Under the background of rapid economic development in this area, increases in anthropogenic aerosol emissions caused by human activities have had a severe impact on the ecological environment. For example, organic aerosol originated from crop burning and coal combustion was reported to have a large component in total aerosol and poses a threat to human health in North India and Southeast Asian Archipelago (Reddington et al. 2014; Singh et al. 2017). The increases in aerosol particles and unfavourable weather conditions were summarized as the reasons of continuity and regional haze in the North China

Plain of China (Che et al. 2014; Huang et al. 2020; Liu et al. 2013). Except for the impact on air quality, the increase in aerosols has been proven to also impact regional climate, delay precipitation and possibly further induce extreme natural disasters including floods and droughts (Guo et al. 2016; Huang et al. 2014a; Rosenfeld et al. 2008). In addition to the overall space-time changes caused by human activities, the existence of some aerosol pollution emergencies can't be ignored, natural sources of aerosol such as smoke, ash plume, and sand storm raised respectively from wildfire, volcano eruption, and

desert of arid region may also undergo physical and chemical reactions in the atmosphere, or be transported to more distant areas of human activity (Eck et al. 2005; Ma and Gong 2012; Zhang et al. 2017). These particles are believed to aggravate regional air pollution and complicate the regional aerosol environment (Horwell and Baxter 2006). These events can also be reflected by the short-term variances of aerosols. In the all, the change of aerosol is an important research topic that deserves long-term attention in Asia.

The developers of satellite technology have been greatly promoted the study of spatial-temporal evolutions and properties of aerosols in the last several decades. Satellite observations are an irreplaceable approach because they can collect aerosol distribution information from space and greatly make up for the limitation of the spatial resolution. Such as sensors like MODerate-resolution Imaging Spectroradiometer (MODIS) and Multi-angle Imaging SpectroRadiometer (MISR), have provided a wealth of aerosol distribution data for more than 20 years (Diner et al. 1998; Masuoka et al. 1998). Benefiting

from the hardware load and channel settings, different sensors have their unique aerosol products: MODIS products can





retrieve Aerosol Optical Depth (AOD) spatial distribution with high accuracy and MISR can further obtain particle size information owing to multi-angle imaging (Sayer et al. 2014). These satellite remote sensing aerosol products described aerosols in the atmosphere from different perspectives and have provided useful aerosol information for related research, including quantifying global transport of particulates, constraining climate models, improving air quality and evaluating

long-term aerosol changes (Lenoble et al. 2013). Although satellite observation has many advantages, verification and calibration satellite aerosol products can't be neglected. Aerosol Robotic Network (AERONET) is the main federated ground-based observation network and public data archive that has continuously measured and stored more than 25 years aerosol characteristic datasets (Dubovik and King 2000; Eck et al. 1999; Holben et al. 1998). The instruments used in this network are sun-photometers, and stations of the network are in major ecosystems and human activity areas around the

world to provide information of concentration, optical, microphysical, and radiative properties for aerosol related research (Holben et al. 2001). Their observation data have successfully used as ground validation reference for multiple satellites (Chen et al. 2020; Levy et al. 2010; Remer et al. 2005).

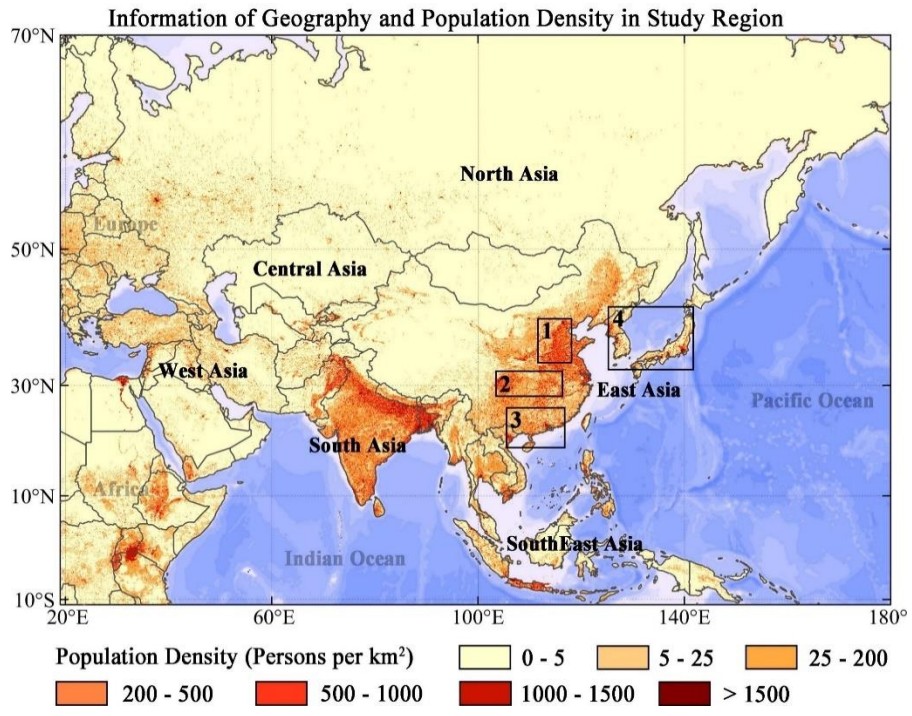

**Figure 1.** National geographic and distribution of population density in Asia. The blue regions represent main ocean systems. East Asia is divided into four different regions shown as the four black boxes in the figure. All regions are divided according to national or provincial boundaries and the details are shown in **Figure S1**.



Current researches on the spatial and seasonal changes of aerosol generally use a certain kind of observational data, such as ground-based sun photometer, lidar, and satellite sensor (Bi et al. 2014; Kumar et al. 2018; Vernier et al. 2011). Among them, satellite-based studies, mostly focused on changes in total AOD, while less attention was paid to the trends of aerosols with different size and types on time and space scales. This was mainly due to the limitations of aerosol sensor and retrieval algorithms in the past. Meanwhile, the ground-based stations are limited by their number which is mostly confined to several locations. Given that the good performance of latest MISR aerosol products (Garay et al. 2020) and the aerosol type analysis ability of the Modern-Era Retrospective analysis for Research and Applications, Version 2 (MERRA-2) is well in Asia (Ali et al. 2022; Ou et al. 2022; Xu et al. 2020), combines of multi-source data (ground-based stations, satellites, and numerical analysis) were expected to make up the limitation of only using satellites AOD products or the ground-based stations in the investigation of the long-term seasonal variability and trends of different properties of aerosols.

In addition, in respect of an in-depth and objective analysis of aerosol, Sayer and Knobelspiesse (2019) recommended that using the geometric mean to aggregate satellite AOD data instead of the arithmetic mean, because aerosol distribution is much closer to the log-normal distribution than the conventionally normal distribution in large and long-term scales. As well as, Collaud Coen et al. (2020) pointed out that the variance corrected Sen's slope and a combined Trend-Free Pre-Whitening (TFPW) Mann–Kendall (MK) test are better to calculate AOD trends and significance. The most important advantage of it is that can avoid the overestimation brought by the interference of aerosol serial auto-correlations and result in an unbiased trend (Wang et al. 2015; Yue et al. 2002). The continuing discussion of the aerosol data and the improvement of analysis methods allow us to re-think the long-term trend of aerosol from a novel perspective. Following the trend analysis method, another novel perspective which considers the small-scale variability of aerosol in a short time or space is presented in this study. For example, when during long-term regional haze, the aerosol loading will be high (Zhao et al. 2013) but the aerosol variability may be relatively low (because AOD has been maintained at a high level). In contrast, high aerosol variability may indicate sudden pollution incidents or temporary emission sources. In addition, for better estimating various aerosol parameters from satellite, in several advanced multi-pixel aerosol retrieval algorithms, temporally and spatially adjacent pixels are considered as priory empirical constraints, and the related algorithms and their result have been well recognized (Dubovik et al. 2021; Lyapustin et al. 2018). However, these algorithms usually constrain the dynamic change range of aerosol properties in the retrieval process or applied several filters to smooth primary aerosol result, therefore the high variability of aerosol may regard as noise and suppressed, and the sudden pollution incidents or temporary emission sources may be neglected, finally. In fact, the constraint from adjacent pixels has been introduced in the field of particle matter concentration estimation for a long time, and it was widely used to give geographical and time weighting factors (He and Huang 2018; Park et al. 2020; Wei et al. 2021). Consequently, in the context of the complex aerosol in Asia, if aerosol retrieval algorithms can consider the high variability of aerosol, they will better quantitatively describe the atmospheric environment.

Therefore, in order to understand the aerosol environment and its evolution in more detail over Asia, a comprehensive investigation (2000-2020) of spatial-temporal distributions, trends, and small-scale variabilities in the different size and type





aerosols was executed in this study, based on multi-source observation data including ground-based stations, satellites, and reanalysis data. The MODIS is used to obtain the total AOD and the MISR and the MERRA-2 are used to acquire aerosol
size and chemical composition information, respectively. The modified TFPW-MK test and Pettitt's test were used to estimate trends, change points, and significance of aerosol parameters, avoiding the interference of serial auto-correlations. In addition, we also evaluated the differences between the main satellite products and ground-based AERONET observations to clarify the possible biases in this study.

## 2 Observations and Data

### 2.1 Study Region

Influenced by a variety of anthropogenic and natural emission sources and monsoons, the Asian aerosol environment is extremely complex, having obvious distributions and trends. Two populous countries: China and India, are widely concerned, and they are also the main anthropogenic emission sources in East and South Asia, respectively. Different from most studies, North Asia (NA) is also considered where aerosol natural sources are dominant, to enrich the understanding of whole Asia.
Therefore, we divide Asia into six major portions, as shown in **Figure 1**: East Asia (EA), South Asia (SA), Southeast Asia (SEA), Central Asia (CA), West Asia (WA), and North Asia (NA), in according with the geography and the boundary line of countries. In addition, the EA is further sub-divided into North China (NC), South China (SC), Central and Western China (CWC), and Korean peninsula and Japan (KJ). The details of division are showed in **Figure S1**. The purpose is to refine the evaluation in regions of EA with high-speed economic development. Finally, for seasonal statistic, the seasons were defined
by month as follows: spring (March, April and May, MAM), summer (June, July, and August, JJA), autumn (September, October and November, SON), and winter (December, January and February, DJF).

### 2.2 Ground-based Observation Sites

Standard aerosol property product (Level 2) was obtained from ground-based sun sky photometer (Cimel Electronique) observations at all available AERONET sites (Holben et al. 1998) in Asia. The AOD was calculated at 8 channels from 340
to 1640 nm with uncertainties less than 0.01 (Eck et al. 1999). Other comprehensive aerosol properties: Single Scattering Albedo (SSA), Absorbing AOD (AAOD), and particle volume size distribution were retrieved at 440, 675, 870, and 1020 nm from the direct sun observations and sky irradiance (Dubovik and King 2000). These products, which have passed the most stringent quality control (Holben et al. 2006), were used to show regional aerosol characteristics and evaluate satellite measurements. In addition, an extra site established at Wuhan University was also applied in this study (Jin et al. 2021). This
site is equipped with the same sun sky photometer as AERONET to supplement the aerosol optical properties in CC area.



### 2.3 Satellite Aerosol Measurements

#### 2.3.1 MODIS C6.1 Aerosol Products

The MODIS sensors carried on the Terra and Aqua satellites were launched in 2000 and 2002, respectively, having been providing continuous earth observations (King et al. 1992). The scanning width of the sensors is ~2300 km with a full
coverage of the world every 1-2 days. Two operational aerosol retrieval algorithms: Dark Target (DT) and Deep Blue (DB) have been applied to produce high-accuracy C6.1 Level 2 aerosol products, with the spatial resolutions of 3 km and 10 km (Hsu et al. 2013; Levy et al. 2013). Among them, the DT algorithm covers areas of the ocean and dense vegetation, while the DB algorithm is only applied over land including both dark and bright surfaces. The current C6.1 version has corrected the overestimations of AOD over urban surface type, by using an improved surface reflectance estimate strategy (Gupta et al.
2016). The MODIS aerosol products are used to obtain AOD parameter in the study, and these products can be acquired from the NASA's Level-1 and Atmosphere Archive (LAADS) Distribution System Distributed Active Archive Center (DAAC).

#### 2.3.2 MISR V003 Aerosol Products

The MISR onboard Terra satellite is a multi-angel sensor with 9 directions between ±70.5° and 4 channels from 443 nm to
865 nm (Diner et al. 1998). Due to the fore-aft viewing strategy, MISR has a narrower swath with relatively high spatial resolutions (0.275-1.1 km), observing the whole world once in ~9 days. In addition to retrieval of total columnar aerosol (Kahn et al. 2005), the multi-angle technology allows MISR to infer size-segregated AOD from total observations: Small-size AOD (< 0.7 um, SAOD), Medium-size AOD (0.7-1.4 um, MAOD), and Large-size AOD (> 1.4 um, LAOD). Compared with the previous version, the latest Level 2 MISR Aerosol parameters Version 3 (V23) product applies two separate
algorithms on land surface and dark water respectively to improve accuracy and resolution (4.4 km) (Garay et al. 2020). The MISR aerosol products are used to obtain size-segregated AOD. The corresponding data can be acquired from MISR: Access Data (nasa.gov).

### 2.4 Reanalysis Data

MERRA-2 is a latest satellite era (1980 onward) retrospective analysis dataset, developed by NASA with Goddard Earth
Observing System version 5, Earth system model (Gelaro et al. 2017). In addition to meteorological observations, MERRA-2 provides several improvements, including AOD assimilation from various ground-based and satellite remote sensing instruments (Randles et al. 2017). It has shown a great performance in AAOD and surface find-mode particles by comparing available satellite, aircraft, and ground-based observations (Buchard et al. 2017). The aerosol species data used in this study was obtained from the MERRA-2 V5.12.4 products (M2TMNXAER), which defined mass concentrate and optical depth of
several aerosol components, including black carbon (BC), dust, sea salt (SS), sulphate, and organic carbon (OC). This product has a spatial resolution of 0.5 °×0.625 ° and is released once a month.





## 2.5 Auxiliary Data

In addition to the main data mentioned above, we also used some auxiliary data to enrich the research content. The global gridded digital elevation data was from the General Bathymetric Chart of the Oceans (GEBCO), the population density data was from the LandScan (Rose et al. 2019), and the active fire counts (confidence > 60%) data was from the MODIS Terra/Aqua Collection 6 fire products (MCD14DL) (Giglio et al. 2016).

## 3 Methodology

### 3.1 Validation of Satellite Aerosol Products against with AERONET

The accuracy of satellite products is evaluated to avoid possible misunderstanding of trends in climatological research due to retrieval deviations. To match satellite and AERONET observations, a common method in aerosol community is applied, which calculated data within a limitation of 30 minutes and a circle of 0.25 ° (~25 km) radius centred on the selected site (Sayer et al. 2013). Measurements of AERONET at 550 nm was calculated by 440 nm and 670 nm following the wavelength depended relationship proposed by Ångstrom (1964). Several indicators were used to evaluate the performances of MODIS and MISR AOD products, including Root Mean Square Error (RMSE), Mean Absolute Error (MAE), Relative Mean Bias (RMB), and the percentage falling into Global Climate Observing System (GCOS) (Popp et al. 2016), as shown in Eq. 1-4. In addition, a linear regression was also used to assess the correlation between AERONET and satellite measurements with a significance (p) test to confirm whether the satellite retrieval result is available.

$$RMSE = \sqrt{\frac{1}{n}\sum_{i=1}^{n}(AOD_i^{Satellite} - AOD_i^{AERONET})^2} \qquad (1)$$

$$MAE = \frac{1}{n}\sum_{i=1}^{n}\left|AOD_i^{Satellite} - AOD_i^{AERONET}\right| \qquad (2)$$

$$RMB = \frac{1}{n}\sum_{i=1}^{n}\left|AOD_i^{Satellite}/AOD_i^{AERONET}\right| \qquad (3)$$

$$GCOS = max(0.04, 0.1 \times AOD_{AERONET}) \qquad (4)$$

It's noting that because MISR and AERONET have different standards for particle size division, direct comparison cannot be performed. A compromise solution is that the SAOD+MAOD of MISR corresponds to the fine-mode AOD of AERONET, and the LAOD of MISR corresponds to the coarse-mode AOD of AERONET, for performance evaluation (Tao et al. 2020). The deviation of the satellite aerosol products will be clarified firstly by comparing the AERONET observations.

### 3.2 Climatology and Long-term Trends Assessment

The long-trend trend assessment of aerosol loadings and different aerosol types is meaningful to improve understandings of local aerosol environment, aerosol emission sources, environmental policy making, health impact related to air quality, and aerosol climate effect. According to different applications, satellite aerosol products required aggregation of the directly observed measurements over different temporal scales, including daily, monthly, seasonal, and annual averages. As Sayer





and Knobelspiesse (2019) reported, aerosols become increasingly more consistent with log-normal than normal distributions for longer time scales such as month and year. Therefore, we used a geometric mean method to re-calculate the monthly data from the Level 2 MODIS C6.1 and MISR V003 aerosol products in accordance with the reported log-normal distribution. It can be expected that this method will restrain contributions from haze events with extremely high AOD and generates lower

average values.

### 3.2.1 Theil–Sen Approach

The Theil–Sen estimator is a robust estimation of the magnitude of a trend proposed by Theil (1950) and Sen (1968). It has been widely used in identifying the trend of linear change in time series data, including studies on hydrological, atmospheric and land (Gui et al. 2021; Hall et al. 2014; Sulis et al. 2011), and is given by the following relation, as Eq. 5.

$$\beta = Median\left(\frac{x_t - x_j}{t - j}\right), \ \forall \ t > j \tag{5}$$

Where, $\beta$ is the slope calculated by Theil–Sen approach. $x_t$ and $x_j$ represent any data in time series that meets the conditions $(t > j)$ in a time series.

### 3.2.2 Modified Trend-Free Pre-Whitening Mann–Kendall Test

The original MK test is a rank-based non-parametric test for assessing the significance of a trend series data (Kendall 1975;

Mann 1945). It is based on the independent and randomly ranked time series, do not require that the trend is stable, and the its results are also not impacted by the presences of outliers or missing value. However, the auto-correlation is usually widespread in time observed AOD series (namely, the data are not strictly independent and random). This makes the original MK test usually results the occurrences of type I error in statistics and overestimates the significance of trends in statistical (Von Storch 1999). For time series of aerosol data, Collaud Coen et al. (2020) presented a combined evaluation method to

reduce the negative effects of auto-correlation and improve the accuracy of trend on different time granularity and segmentation. Specifically, the trend of aerosol is determined by the variance corrected Sen's slope (Wang et al. 2015) and the trend is considered statistically significant only when both pre-whitening (Kulkarni and Von Storch 1995) and TFPW (Yue et al. 2002) series can pass the MK tests. However, a disadvantage of pre-whitening is that it removes a part of the trend while removing the autocorrelation and resulted in a lower sensitivity. This will cause a part of the aerosol trend to be

lost during processing. Taking the monthly MODIS AOD as an example, **Figure S2** shows the serial auto-correlation using different data processing methods. It can be found that compared with original AOD series, lag-1 serial coefficient of both pre-whitening and TFPW series show a similarly pronounced decrease. This means that the auto-correlation has been well suppressed in TFPW series for the current data set. Therefore, in order to preserve more details of aerosol changes and avoid overestimation of trends in this study, the trend and significance were determined by the variance corrected Sen's slope and

TFPW-MK method, respectively. The specific steps are as follows:





**Step 1.** For original AOD series ($X_{t,T}$), calculate the lag-1 serial coefficient to determine whether this series has significant serial auto-correlation, as Eq.6.

$$r_1^o = \frac{(n-1)^{-1}\sum_{t=1}^{T-1}[X_{t,T}-E(X_{t,T})][X_{t+1,T}-E(X_{t,T})]}{(n)^{-1}\sum_{t=1}^{T}[X_{t,T}-E(X_{t,T})]^2} \tag{6}$$

Where, the $r_1^o$ is the lag-1 serial coefficient for original AOD series $X_{t,T}$ and the $E(X_{t,T})$ is the mathematical expectation. If

the $r_1^o$ do not fall into the significant intervals using the two-tailed test (Yue et al. 2002), the series $X_{t,T}$ is considered serially uncorrelated that can use the MK test directly. The $t$ represents the time position of data in the series and the $T$ is the length of data (namely, there is $t = 1, 2, 3, …, T$).

**Step 2.** If the $X_{t,T}$ is serial auto-correlated, calculate the Sen's slope ($\beta$), remove the serial trend, and create the blended series, as Eq. 7-9.

$$X'_{t,T} = X_{t,T} - \beta t \tag{7}$$

$$Y'_{t,T} = X'_{t,T} - r_1^D X'_{t-1,T} \tag{8}$$

$$Y_{t,T} = Y'_{t,T} + \beta t \tag{9}$$

Where, the $X'_{t,T}$ is the de-trended series, the $Y'_{t,T}$ is the residual series (de-trended and lag-one autoregressive removed), and the $Y_{t,T}$ is the blended series (mixing residuals and trends). The $r_1^D$ is the lag-1 serial coefficient of $X'_{t,T}$, calculated following

Eq. 6. Then, the significance of trend and abrupt change points are assessed from the blended series. It should be noted that the influence of tied rank groups was ignored here in the calculation of variance in the MK test, because the values of AOD observed from satellite were usually different.

**Step 3.** The unbiases trend estimation ($\beta'$) can be approximately estimated by $\beta$, as Eq. 10.

$$\beta' = \beta\sqrt{(1+r_1^D)/(1-r_1^D)} \tag{10}$$

**3.2.3 Change Point Test of AOD Series**

Analysis of change point is a technology in accordance with the homogeneity testing in a time series (Alexandersson and Moberg 1997). The technology determines the time point at which the interruption or change point occurred, and thus, in study of long-term aerosol parameter observation, it can help to reveal different patterns of aerosol changes closely related to human activity. Similar to the MK test, the Pettitt test (Pettitt 1979) is also a non-parametric test, based on the ranks of series

data. The advantages of Pettitt test are that it is sensitive to breakpoints in the middle of a time series and robust against outliers and data with skewed probability distributions (Wijngaard et al. 2003). Moreover, compared with the MK change point test, the Pettitt change point test only reflects the most important change and avoids to find multiple unstable patterns when trends or breakpoints are not significant in the time series data. Here, the calculation of Pettitt change point is as follow:

$$U_{t,T} = U_{t-1,T} + V_t \tag{11}$$

where, the $U_{t,T}$ is a rank statistic series in the Pettitt test and the $V_t$ is referring to the rank at position of $t$. For $t > 2$, there is

$$V_t = \sum_{j=1}^{T} sgn(y_t - y_j) \tag{12}$$





The $y_t$ and $y_j$ are specific values in the blended series $\boldsymbol{Y}_{t,T}$ and the $sgn$ is a symbolic calculation consistent with the definition in the MK test. When $t = 1$, there is $\boldsymbol{U}_{t,T} = V_t$ ($U_{1,T} = V_1$). Then, the Pettitt change point ($K_T$) is find at the position with maximum absolute value of $\boldsymbol{U}_{t,T}$,

$$K_T = \max_{1 \leq t \leq T} \left| \boldsymbol{U}_{t,T} \right| \tag{13}$$

The significance probabilities ($p$) associated with the Pettitt change point ($K_T$) can be approximately estimated by

$$p = 2 \times \exp\left[\frac{-6 \times (K_T)^2}{(T^3 + T^2)}\right] \tag{14}$$

### 3.2.4 Percent Change Calculation

Since the magnitude of different aerosol parameters can be very different, we also calculated the percentage of change during the whole study period to make the trend comparable between these parameters. In accordance with the study of Yue and Hashino (2003), the percentage of change is calculated from the trend and time span, following Eq. 15.

$$Percent\ Change\ (\%) = \frac{\beta' \times length\ of\ Year}{Mean} \times 100\% \tag{15}$$

The $\beta'$ is the variance corrected Sen's slope as the obtained trend and the $Mean$ represents regional geometry average values of different aerosol parameters. Here, we evaluated aerosol trends from long-term data of different observation systems at two different levels, the pixel level and the region-level.

### 3.3 Estimation of Spatial and Temporal Small-scale AOD Variability

The small-scale variability of AOD in the study refers to the degree of aerosol oscillation on a small temporal and spatial scale, and can be used to clarify the stability of aerosol loading at a certain place. For example, a region with a high AOD variability means that aerosol can change drastically at certain moments, probably due to the influence of abrupt emission events, such as long-distance transported dust, air pollutant migration and wildfire. A low AOD variability, in contrast, generally reveals a relatively stable source of aerosol emissions or aerosol environment in the local. In addition, several advanced aerosol retrieval algorithms also start to introduce the spatiotemporal change information of aerosols through multi-pixel synergetic retrieval or filtering as constraint or additional observation. Therefore, the small-scale AOD variability can supplement the characteristics of aerosol from another perspective, and is meaningful for aerosol research. In previous studies, standard deviation is a useful and common index to indicate the degree of dispersion of the data. In climate-related research, it can be used to partially characterize temperature distributions, extreme precipitation probabilities, or model accuracy (Boer 2009; Gu et al. 2019; Unkašević et al. 2004). In this study, the small-scale variability of AOD is also defined as the standard deviation of AOD over a small period or over a small range of space. Thus, it can be easily calculated from different satellite products. Sliding windows were used to calculate the standard deviation, with scales of 3x3 pixels for spatial and 7 days for temporal, respectively.





$$STD = \sqrt{\frac{\sum_{i=1}^{N}(AOD_i - \overline{AOD})^2}{N-1}}$$

$\qquad$ (16)

Where, $N$ is the number of sample (9 for spatial and 7 for temporal) and $\overline{AOD}$ is the numerical average value of AOD in a small-scale. Missing data will be removed. Like the other aerosol parameter, the small-scale variability of AOD was also used to estimate the distribution as an additional statistical aerosol parameter. Through the geometric mean method and the

300 small-scale variability, it can be expected to explore aerosol distribution and variation characteristics from more perspectives.

## 4 Results and Discussions

### 4.1 Performance of MODIS and MISR Aerosol Products

Before long-term analyses, we assessed the performance of satellite aerosol products used in this study firstly to avoid potential misunderstanding of trends caused by retrieval deviations, especially in Asia where the aerosol loading can vary

greatly. **Table 1** showed quantitative indexes of retrieval performance of AOD products from MODIS and AOD, SAOD, MAOD, LAOD and AAOD products from MISR. It was found that the GCOS percent was higher than 40% and the R was higher than 0.87 for all MODIS AOD products. The MAE is within 0.10 and the RMSE was floating around 0.14, showing a good performance. Considering that MODIS DT and DB have their own advantages, and their combination can increase the coverage area while maintaining performance. For AOD parameter, the DT&DB were selected for trend analysis of AOD as

a balance of spatial coverage and accuracy.

**Table 1.** Validations of key aerosol parameters from MODIS and MISR by compared with AERONET observations.

|  | MAE | RMB | RMSE | GCOS | Slope | Intercept | R |
|---|---|---|---|---|---|---|---|
| *MODIS/Terra* |  |  |  |  |  |  |  |
| DT | 0.080 | 1.308 | 0.133 | 43.193 % | 1.019 | 0.035 | 0.909 |
| DB | 0.090 | 1.160 | 0.140 | 40.289 % | 0.902 | 0.030 | 0.872 |
| DT&DB | 0.087 | 1.351 | 0.141 | 40.901 % | 0.994 | 0.041 | 0.893 |
| *MODIS/Aqua* |  |  |  |  |  |  |  |
| DT | 0.080 | 1.195 | 0.136 | 45.646 % | 0.980 | 0.032 | 0.896 |
| DB | 0.090 | 1.169 | 0.141 | 41.284 % | 0.870 | 0.039 | 0.871 |
| DT&DB | 0.086 | 1.255 | 0.142 | 43.078 % | 0.955 | 0.040 | 0.884 |
| *MISR/Terra* |  |  |  |  |  |  |  |
| AOD | 0.071 | 1.181 | 0.152 | 54.194 % | 0.597 | 0.088 | 0.863 |
| S+MAOD | 0.056 | 0.985 | 0.015 | 65.130 % | 0.425 | 0.043 | 0.842 |
| LAOD | 0.030 | 2.836 | 0.004 | 79.301 % | 0.315 | 0.037 | 0.480 |



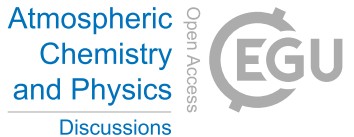

| | | | | | | | |
|---|---|---|---|---|---|---|---|
| AAOD | 0.037 | 0.269 | 0.003 | 61.137 % | -0.004 | 0.007 | -0.023 |

*Note.* DT means AOD is retrieved from the dark target algorithm, DB means AOD is retrieved from the deep blue algorithm, and DT&DB represents combined AOD from dark target and deep blue algorithm. The S+MAOD is the sum of the SAOD and MAOD. Linear regressions of AAOD do not pass the significance test with the p values of 0.735. This means that the AAOD cannot participate in the trend assessment. All other parameters passed the significance test (p < 0.05).

Compared with MODIS AOD products, the MISR AOD showed a higher GCOS percent (54.194 %) but lower R (0.863) and slope (0.597). The high GCOS percent can be attributed to the multi-angle information, making for more stable retrieval and fewer outliers. However, due to decreasing spectral contrast, high aerosol loading conditions will reduce the sensitivity of the MISR retrieval method to separate surface reflections from multi-angle information (Limbacher et al. 2022) and thus resulted to underestimate the value of AOD (Fan et al. 2023). For size-segregated AOD, the S+MAOD from MISR showed a high R (~0.868), meaning that it can be used to obtain information about fine-mode aerosol. However, since they inherited AOD values, the underestimation still existed, which might lead to the estimated trend being less than the actual change. In addition, for LAOD, the RMB was larger than 2 and the R was very low (0.480), revealing the LAOD had a large positive deviation in the low value area. Nevertheless, the LAOD passed the significance test of linear regression and thus we can still use LAOD data in trend analysis. On the contrary, the AAOD cannot pass the significance test and it was removed from this study.

Probability density functions were also calculated at three AOD intervals as shown in **Figure 2**. When the values of AOD were less than 0.25, AOD products from both MISR and MODIS DT&DB dataset showed slight overestimations (the median bias of ~0.02). Then, with the AOD increasing, the slight overestimations were maintained for Terra AOD products; whereas an underestimation (~-0.03) appeared in Aqua AOD products under the condition of AOD > 0.75. By using the average values of the two, this opposite bias can be expected to be partially offset in the overall distribution. However, for MISR, the median biases were ~-0.04 and ~-0.3 when AOD was between 0.25 and 0.75 and larger than 0.75, respectively. This underestimation discussed above, which increased with the increase in AOD, also appears in the size-segregated AOD products of MISR and must be clarified and noted.

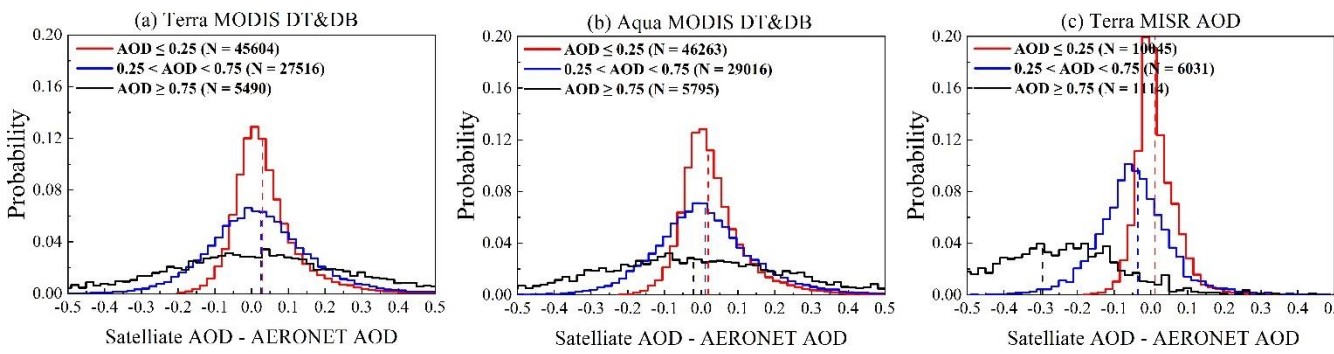





**Figure 2.** Probability distributions of the difference between Satellite ((**a**) Terra MODIS, (**b**) Aqua MODIS, and (**c**) MISR)
and AERONET AOD products. The vertical dashed lines are in the median position of each sample group.

**4.2 Distribution Characteristics of Different Size and Type Aerosols in Asia**

In this section, we studied the spatial distributions of aerosols with different size and type. The spatial distribution of AOD
was closely related to population density, and most of the land was dominated by the small particles (SAOD). As shown in
**Figure 3**, the spatial distribution characteristics of multiyear mean size-segregated AOD were investigated. In Asia average
terms, the values of AOD, SAOD, MAOD, and LAOD were 0.153, 0.074, 0.007, and 0.033, respectively. The AOD was
highest in SA (0.291), followed by WA (0.234), SEA (0.178), EA (0.144), and CA (0.128) and the lowest in NA (0.110).
Although the mean AOD in EA is not high, heavy aerosol loadings were concentrated in the regions of China with the AOD
values of 0.419, 0.214, and 0.368 in NC, CWC, and SC, respectively. Spatial distribution of SAOD has a high consistency
with AOD, which is because aerosols on land are mainly composed of small particles (~60%). The highest SAOD value was
350 in SA (0.135). Whereas, due to the low population density, emissions of fine particle caused by human activities were also
the lowest in NA with SAOD value of 0.052. In contrast, the value of MAOD was the lowest in all size-segregated AOD
showing the weakest contribution of extinction, owing to their narrow particle size range. The distribution of MAOD was
mainly in regions of sea and coast, indicating that this type of aerosol was possibly the SS particles. While, the LAOD was
mainly distributed in the arid regions of Western China, Northern SA, CA, and the Arabian Peninsula, indicating dust
aerosol type. The remaining SAOD mainly come from small particles, such as sulphate, OC, and BC particles in accordance
with MERRA-2 data.

Seasonal characteristics of size-segregated AOD were shown in **Figure 4**. From the whole land region of Asia, the highest
AOD was found in spring (0.190), followed by winter (0.159), summer (0.152), and autumn (0.117). Here, it is noted that
due to snow cover, there was a large number of missing retrievals in winter of NA, which would lead to an overestimated
mean AOD for the whole Asia. The various changes of aerosol distributions in different seasons are mainly caused by
atmospheric circulations, local meteorological conditions, and intensity of regional anthropogenic and natural emissions.
Due to the large proportion of small-size particle in the aerosol, the SAOD and total AOD have a high consistency. The
highest value of SAOD was in spring (0.094), followed by summer (0.076), and is lowest in autumn (0.059). Among them,
changes of SAOD in SEA and NA were characteristic. High values of SAOD were concentrated in the Indochina peninsula
in the spring while Indonesia in Autumn. This distinct difference in seasonal distribution suggests a significant impact of
human activities on aerosols in SEA. And high SAOD of NA was found in spring and summer, implying events of wildfire,
considering the sparse people population there. By contrast, changes of MAOD and LAOD were generally related to dust
event originating from arid and semi-arid regions, including Arabian Peninsula, Northwest SA, and Northwest China. The
dust particles can widely transmit with the atmospheric circulation (Ma et al. 2015; Zhu et al. 2007) and they can reach other
land and sea surfaces to deposit or participate in cloud formation, further affecting regional climate and air quality (Husar et
al. 2001; Wang et al. 2021). The most typical particle transport phenomenon in Asia was found in spring of the northern





Pacific Ocean. The migration of these particles resulted in significant increases in AOD at different scales in that region, in sharp contrast to the clean ocean surface. The most obvious was the increase of MAOD, suggesting that medium-sized dust particles. Meanwhile, an obvious incident of large particle pollution was found in the northern Indian Ocean in summer, with the percent of MAOD+LAOD over 50%. This phenomenon may be caused by the surge of dust transport and SS aerosols under favourable weather conditions such as an increase in ocean wind speed. To summarize, the seasonal cycle of particles by different size was of great significance for understanding the characteristic and change of aerosols.

**Figure 3**. Spatial distributions of annual average of **(a)** AOD, **(b)** SAOD, **(c)** MAOD, and **(d)** LAOD in Asia from 2000 to 2020. The total AOD was from MODIS C6.1 DT&DB dataset and the size-segregated AOD was from MISR V23 aerosol products.



**Figure 4.** Spatial distributions of seasonal AOD, SAOD, MAOD, and LAOD obtained from MODIS and MISR aerosol products in Asia from 2000 to 2020. The AOD was from MODIS C6.1 DT&DB dataset and the size-segregated AOD was from MISR V23 aerosol products.

The type of aerosol particles can largely reflect their chemical composition and source. Here, to further study aerosol properties in Asia, we calculated average distributions of sulphate, dust, SS, OC, and BC AOD from MERRA-2 reanalysis data, as shown in **Figure 5**. The differences in the distribution of type-segregated AOD provide support for the reason of the distribution and change in size-segregated AOD. Generally, the impact of dust aerosols on the Asia showed the apparent dependence of the seasons, which was consistent with LAOD observed from MISR. The values of dust AOD were highest in





spring (0.059), followed by summer (0.036), autumn (0.023), and winter (0.022). The WA was the most severely affected area by dust type aerosols with the mean value of dust AOD of ~0.142 and showed a clear seasonal cycle: the dust AOD in spring (0.199) and summer (0.180) was almost the double that in autumn (0.107) and winter (0.081) in this area. By contrast, NA was rarely affected by dust aerosol that the values of LAOD (0.013) and dust AOD (0.020) were low and 76.4% of AOD is contributed by SAOD. The sulphate AOD did not change significantly in a seasonal cycle in Asia. The highest value was found in SA (0.113) and followed by EA (0.086). This is closely related to intensive human activity that a large number of aerosol particles originate from the gas-to-particle conversion during the rapid progresses of industrial developments (Mallet et al. 2003). Similarly, the seasonal distribution of SS aerosols was also relatively stable, with SS AOD values of ~ 0.003 in a whole study area. Whereas, an abnormal high value of SS AOD (> 0.15) was found in the Northern Indian Ocean in summer, suggesting a change in aerosol types. This means, in addition to the dust aerosol, the SS aerosol is also an important natural source causing the large values of MAOD and LAOD in this region. Moreover, previous studies show that this phenomenon is influenced by the monsoon and higher concentration of SS aerosol can also promote the formation of sulphate aerosol (Alexander et al. 2005; Satheesh and Srinivasan 2002).

Carbonaceous aerosol is a general term for a series of carbon-containing aerosols. Although its concentration is relatively low in the atmosphere, it has profound effects on the entire climate system (Ramanathan and Carmichael 2008). The main source of the carbonaceous aerosol is biomass and fossil fuel combustion and automobile exhaust emission and can also be produced in the processes of the secondary organic rich and conversion (Huang et al. 2014b). In Asia, the OC AOD showed obvious temporal and spatial distribution differences that the regional average values were 0.029 in spring, 0.036 in summer, 0.019 in autumn, and 0.014 in winter, respectively. And the contents of BC were lower than that of OC, with 0.006 in spring, 0.008 in summer, 0.009 in autumn, and 0.007 in winter, respectively. From the view of spatial distribution, high values of OC and BC AOD were mainly found in regions with dense population, such as SEA, SA, and EA. This phenomenon is related to biomass burning of the crop residues, which will release carbonaceous aerosols that spread to surrounding areas with turbulence and have profound effects on the regional climate (Ding et al. 2021; Jethva et al. 2019). Especially in SEA, the biomass burning resulted in two distinct patterns of distributions of OC and BC AOD, that is, the BC and OC aerosols concentrated around the Indo-China Peninsula in winter and spring while around Indonesia in summer and autumn. In addition to this, nature wildfires are the major emission sources of carbonaceous aerosols in NA, resulting in higher average values of OC (0.072) and BC (0.009) AOD in summer. This distinctive characteristic is well reflected and confirmed in the fire count data from MODIS observations, as shown in **Figure S3**.







**Figure 5.** Spatial distributions of seasonally sulphate, dust, SS, OC, and BC AOD obtained from MERRA-2 reanalysis data in Asia from 2000 to 2020.





To further estimate composition of aerosols in different regions, percentages of size-segregated and type-segregated AOD in the total AOD were shown in **Figure 6.** The percentages of SAOD, MAOD, and LAOD were calculated from the MISR observations while the percentages of sulphate, dust, SS, OC, and BC AOD were calculated from the MERRA-2 reanalysis data, respectively. The size-segregated AOD fractions showed a consistent pattern in the study area that the SAOD (52.8% - 76.4%) contributed highly to the total AOD comparatively to the MAOD (3.8% - 12.0%) and LAOD (19.7% - 37.6%). The average percentage of SAOD fraction was highest in NA (76.4%), followed by SEA (67.3%), and lower in SA (55.5%) and WA (52.8%). Given that NA area has very low population coverage, the high SAOD fraction there is owing to the atmospheric background aerosol and biomass emission, without source of large particle such as dust aerosol in the desert. By contrast, the highest percentage of MAOD (12.0%) was found in SA and the highest percentage of LAOD (37.6%) was found in WA, showing the influence of dust and SS aerosol. In the meantime, the relative frequency of SAOD and LAOD fractions also showed larger overlaps in SA and WA, suggesting that the origin of the two size aerosol particles probably had the large seasonal variations. From distributions of percentages, this phenomenon can be owing to that monsoon can cause massive concentrations of large particles in the Arabian Gulf coast and thus change particle percentages in the surrounding areas in the summer (Mhawish et al. 2021).

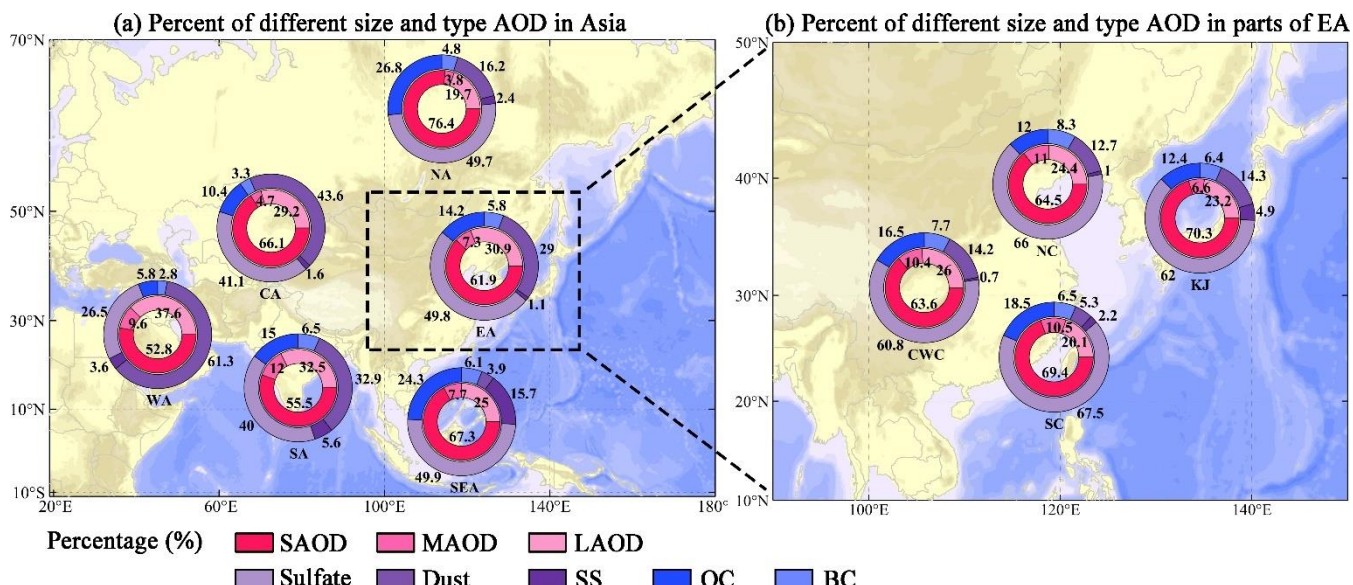

**Figure 6.** Percentage of size-segregated AOD (SAOD, MAOD, and LAOD) and type-segregated AOD (sulphate, dust, SS, OC, and BC AOD) in the total AOD over **(a)** Asia and **(b)** parts of EA.

Complex aerosol characteristic in Asia is derived from multiple different emission sources and unique monsoon behaviour (Li et al. 2016). To further illustrate the differences of aerosols, the type-segregated AOD percentages were also calculated. It was found that, for the whole Asia, the sulphate (50.0%) aerosol is predominant, followed by the dust (24.8%), OC



(17.4%), and BC (5.2%), and the SS (2.6%) aerosol showed the lowest percentage. Among them, influenced by sources of dust in arid and semi-arid areas, the dust AOD in WA and CA is higher than other type-segregated AOD, with the percentages of 61.3% and 43.6%, respectively. It needs to be pointed out that MERRA-2 reanalysis data will overestimate in dust sources in EA and underestimate in anthropogenic dust sources and remote regions (Han et al. 2022). Therefore, satellite data still show that these two regions are dominated by SAOD. The similar high dust AOD was also found in SA

(32.9%). However, although it is greatly affected by dust aerosol, the SA area is still dominated by fine-mode particles due to intense anthropogenic emissions, such as sulphate, OC and BC (Paliwal et al. 2016). For OC aerosol, the highest percentage appeared in NA (26.8%) and followed by SEA (24.3%). This is in accordance with extensive biomass burning events mentioned above. By contrast, the BC aerosol is more likely to be produced in fossil fuel combustion, and thus in addition to biomass burning, BC is also high in human-intensive areas, such as SA (6.5%) and EA (5.8%). The sulphate

aerosols are mainly derived from sulphides emitted in industrial production, and both SEA, EA, and NA have a similar high sulphate percentage with the value from 49.7 to 49.9%. Changes in aerosol type can be reflected in changes in ability of absorbing and scattering, as shown in **Figure S4,** which displayed AAOD and SSA observed by AEROENT sun sky photometers. Aerosols from WA have the highest scattering ability (SSA > 0.96), corresponding to clean desert dust (Costa et al. 2006). By contrast, the high AAOD appeared in SA, SEA, and Central China, with values from ~0.03 to ~0.06.

**4.3 Long-term Trends of Different Size and Type Aerosols**

To further study evolutions of aerosol in Asia, the trends and changes of size-segregated and type-segregated aerosol were described on a spatial scale by using the TFPW-MK method and Pettitt change point test. The rate of change was calculated based on the variance corrected Sen's slope, with units of each year. **Figure 7** showed the trends and change points of AOD in different areas of Asia from observations of MODIS. Among them, significant increase trends in AOD were found in SA

and NA, while a significant decrease trend was found only in EA. The largest increase of AOD ($1.25 \times 10^{-3}$ per year) was in SA, imply that impact of increases in human activity and anthropogenic emissions in densely populated areas. Here, it must be clarified that the AOD trend reported in **Figure 7** seems to be obviously less than that in other recent studies (Gupta et al. 2022; Huang et al. 2021; Mhawish et al. 2021). On the one hand, this is because the methods of geometric mean and variance corrected Sen's slope completely removes the residual auto-correlation component of the trend while reducing the

weight of high AOD value in the air pollution event. On the other hand, trends within different pixels over a large range may be opposite and will cancel out statistically. Such as in SA, the AOD showed highly increasing trends in central and southern India, but it had decreasing trends around Pakistan. By contrast, the increasing trend of AOD found in NA ($4.29 \times 10^{-4}$ per year) was more probably due to biomass emissions, since wildfires are a major local source of aerosols. The AOD change in EA was notable as this was the only area where a decrease ($-5.28 \times 10^{-4}$ per year) in AOD was observed in Asia. Particularly,

the significant decreases in AOD were observed in each sub-region of EA (**Figure 7g-j**). The fastest decrease was found in SC ($-5.24 \times 10^{-3}$ per year), followed by CWC ($-2.64 \times 10^{-3}$ per year) and NC ($-1.71 \times 10^{-3}$ per year), and the slowest was in KJ ($-8.00 \times 10^{-4}$ per year). This meant that the decrease in aerosol loadings in EA and even in Asia is mainly contributed by China.





This phenomenon that rapid decrease in AOD over China has been widely reported (Shi et al. 2020; Xie et al. 2019; Zhang et al. 2018) and it is mostly attributed to a series of effective emission reduction measures, such as dust removal in industrial

production, coal fired power plants denitrification, and restrictions on straw burning (Mao et al. 2014). The change point showed that the responses of AOD to the policy-based emission reduction appeared in 2014, 2012, and 2014 in NC, CWC, and SC, respectively. Whereas, before that, due to rapid economic development, China's aerosols showed an upward trend from 2000 to 2010 (Luo et al. 2014). The trends before and after change points can be found in **Table S1,** as references.

Distributions of Annual AOD and Change Points from MODIS Observations





**Figure 7.** Annual AOD distributions and change points calculated by TFPW-MK and Pettitt methods, respectively, based on MODIS DT&DB observations. The box plots represent AOD and the red curve lines are the rank statistics of AOD as defined in Pettitt change point test. The black dotted lines show two-tailed significance range. If the maximum absolute value of the rank statistic is out of the range, it means that the change point is significant (p < 0.05) and occurs at the time as a red vertical line shown. The significances of AOD trends are indicated by asterisks: * p < 0.05.

**Figure 8.** Long-term trends of seasonal AOD, SAOD, MAOD, and LAOD in Asia from 2000 to 2020. The total AOD is from MODIS C6.1 DT&DB dataset and the size-segregated AOD is from MISR V23 aerosol products. All the points shown in the figure have passed the significance test (p < 0.05). Gray colour represents that the trend is not significant.

To further investigate the changes of aerosol in Asia, spatial distributions of long-term trend of AOD, SAOD, MAOD, and LAOD were calculated during different seasons and shown in **Figure 8**. The both AOD, SAOD, MAOD, and LAOD





exhibited obviously positive trends over most region of SA during spring, autumn, and winter. The summer increase trend of
AOD was the weakest in SA and the main source of contribution was SAOD. This can be attributed to the action of the
southwest monsoon in summer (Li et al. 2016), which brings dust and SS aerosols, and takes away the increasing number of
fine particles emitted by humans in the meantime. The obvious positive trend of AOD in summer was only found in NA,
which was in accordance with the wildfire events. In addition, some regions in WA and CA also showed significant positive
trends of AOD. This is mainly caused by the increase of larger aerosol particles. As mentioned, the reduction of aerosols in
EA is a phenomenon that is worthy to get attention. The largest decrease in AOD of EA was in the spring ($-2.98\times10^{-3}$ per
year) and followed by summer ($-1.29\times10^{-3}$ per year), while no significant decrease in AOD was found in autumn and winter.
Further, compared with other season, the decrease in AOD was also accompanied by widespread decreases of SAOD, MOD,
and LAOD in spring of EA, revealing that the fine particles were reduced while the transport of dust was well controlled.
More specifically, the pronounced reduction in SAOD was CWC ($-1.12\times10^{-3}$ per year) and SC ($-2.53\times10^{-3}$ per year). And the
regions with the obvious reductions of large particle size were the NC and SC regions, with the LAOD trends of $-3.70\times10^{-4}$
per year and $-3.87\times10^{-4}$ per year, respectively. This means that the eastward transportation of spring dust has been greatly
suppressed in recent years in China. By contrast, the decrease in AOD of EA was mainly only due to the small size particles
(SAOD) in summer, which contributed to ~90% of the total AOD decrease. In addition, the trend estimated from MISR is
generally smaller than the real value, which is due to the underestimation of AOD by MISR operational aerosol products
under heavy loading aerosol conditions.

The long-term trends of different aerosol types partially explain and enhance the understanding of the regional aerosol
source changes. As shown in **Figure 9**, the trends of five types (sulphate, dust, SS, OC, and BC) of AOD were estimated by
using MERRA-2 reanalysis data from 2000 to 2020. Comparing with satellite aerosol product, trends calculated by MERRA-
2 product are more consistent. It was found that the largely positive trend of AOD in SA was mainly from increasing
sulphate ($1.09\times10^{-3}$ per year), OC ($2.68\times10^{-4}$ per year), and BC ($5.38\times10^{-5}$ per year). And the positive trend of AOD in NA
was mainly owing to OC ($2.71\times10^{-4}$ per year), and BC ($1.92\times10^{-5}$ per year). These findings confirmed the previous
speculation on the increases in AOD, which are derived from industrial development and anthropogenic emission in SA and
from widely wildfire in NA, respectively. By contrast, the negative trend of AOD in EA was observed with decreasing dust,
SS, BC aerosols. More specifically, a same pattern was found in all NC, CWC, and SC that sulphate AOD increased and
other (dust, SS, OC, and BC) AOD decreased. Whereas, in KJ, the sulphate AOD was also found to decrease while the
change in OC was not significant. The trends of size-segregated and type-segregated AOD were shown in **Table 2** in detail.
In general, from the perspective of statistic in Asia, the sulphate ($-1.04\times10^{-4}$ per year), dust ($-9.22\times10^{-5}$ per year), and SS ($-1.69\times10^{-5}$ per year) AOD decreased while OC ($2.00\times10^{-4}$ per year) and BC ($2.20\times10^{-5}$ per year) AOD increased. The growth
of carbonaceous aerosols such as OC and BC become a feature of the Asian region. However, it is particularly important to
note that in the context of the growth of carbonaceous aerosols such as OC and BC, only the region of China has decreased
in OC and BC.





Sen's slope of type-segregated AOD ($\times 10^{-3}$ per year)

<2.5 -2.0 -1.5 -1.0 -0.5 0.0 +0.5 +1.0 +1.5 +2.0 >+2.5

**Figure 9.** Long-term trends of seasonal sulphate, dust, SS, OC, and BC AOD in Asia from 2000 to 2020. These type-segregated AODs are obtained from MERRA-2 reanalysis data. All the points have passed the significance test ($p < 0.05$). Gray colour represents that the trend is not significant.

**Figure 10** showed the percentage of change in different sizes and types AOD from 2000 to 2020, which was used to study the extent of changes in the aerosol environment in regions of Asia. Compared with the value of trends, the percentage can more intuitively show the change result of aerosol. For Asia region, the percentage of SAOD (-3.34%), sulphate AOD (-





3.07%), dust AOD (-5.51%), and SS AOD (-9.80%) decreased, whereas the percentage of OC AOD (17.09%), and BC AOD (6.23%) increased. The increases in OC and BC were notable in Asia, and this phenomenon was mainly and specifically owing to the increases in OC and BC in SA (13.25% and 6.19%), WA (37.45% and 11.40%), CA (35.64% and 13.30%), and NA (13.47% and 6.62%), respectively. SA was the region with the most consistent change in aerosols, with almost all types

of aerosols showing increasing trends. Whereas, SEA was the region with the least change in aerosols, with only a decrease in the SS AOD (-4.78%). However, the different aerosols in EA were almost all showing downward trends. Although increases in sulphate AOD were also observed in NC (7.06%), CWC (8.18%), and SC (6.84%), due to the offset from other types of aerosols, the total AOD was still on a downward trend in general. Among them, the percentage of OC AOD decreased the most in CWC (-6.39%), while the percentage of BC AOD decreased the most in SC (-9.11%). The above

analyses reveal the long-term changes in size-segregated and type-segregated AOD from different perspectives, and in particular expose a strong contrast, that is, OC and BC AOD decrease in EA but increase widely in other regions of Asia from 2000 to 2020. Specific details are shown in **Table 2**.

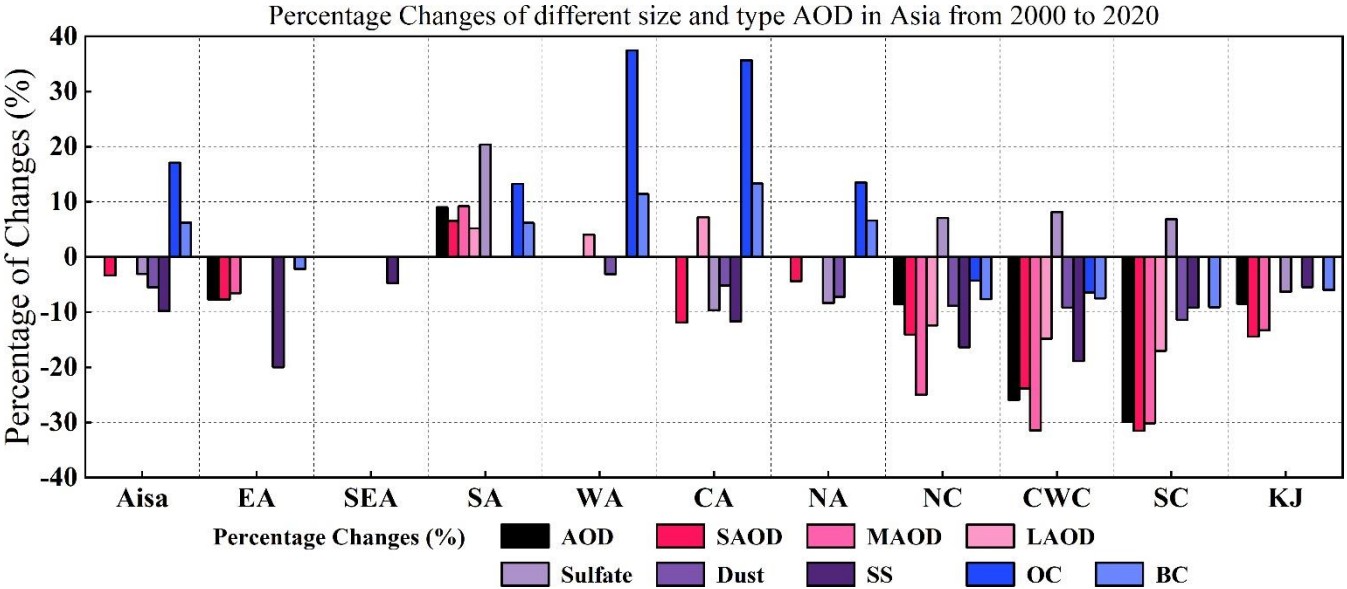

**Figure 10.** Total percent changes in different size and type AOD from 2000 to 2020. Only trends that pass the significance test ($p < 0.05$) are displayed in this figure.

**Table 2.** Change percentage (%) of AOD with different size and type in regions of Asia from 2000 to 2020

| Region | SAOD | MAOD | LAOD | Sulphate | Dust | SS | OC | BC |
|--------|------|------|------|----------|------|------|------|------|
| **Asia** | -3.34 | | | -3.07 | -5.51 | -9.80 | 17.09 | 6.23 |
| **EA** | -7.76 | -6.56 | | | | -19.92 | | -2.19 |





| | | | | | | | | |
|---|---|---|---|---|---|---|---|---|
| **SEA** | | | | | | -4.87 | | |
| **SA** | 6.57 | 9.18 | 5.17 | 20.37 | | | 13.25 | 6.19 |
| **WA** | | | 4.04 | | -3.12 | | 37.45 | 11.40 |
| **CA** | -11.90 | | 7.22 | -9.66 | -5.18 | -11.7 | 35.64 | 13.30 |
| **NA** | -4.37 | | | -8.36 | -7.28 | | 13.47 | 6.62 |
| **NC** | -14.14 | -25.01 | -12.42 | 7.06 | -8.88 | -16.4 | -4.31 | -7.62 |
| **CWC** | -23.84 | -31.47 | -14.82 | 8.18 | -9.18 | | -6.39 | -7.51 |
| **SC** | -31.54 | -30.14 | -17.05 | 6.84 | -11.42 | -9.21 | -18.90 | -9.11 |
| **KJ** | -14.47 | -13.30 | | -6.32 | | | -5.48 | -6.03 |

Note: Only trends that pass the significance test ($p < 0.05$) are shown.

### 4.4 Variabilities of AOD in Small Temporal and Spatial Scales

The long-term analysis reveals the spatial distributions and the time-series changes of different size and type aerosols, which can help to understand the aerosol evolution over these two decades. The small-scale variability shows the changes of aerosols in a short period of time or space, which can reflect the stability of local aerosol emission and identify sudden pollution events, and compensate for the suppression of heavy pollution events by the geometric mean method. From the perspective of aerosol retrieval, the variability of aerosol can also act as a constraint on the multi-pixel inversion algorithm, which was not widely considered and discussed. **Figure 11** showed the temporal and spatial small-scale variabilities of AOD (standard deviations). It must be pointed out that the study of small-scale variability of AOD is based more on the significance of the statistic than physics. It was found that the calculated temporal variability was much greater than the spatial variability, since the aerosols are usually more uniform in spatial distribution. But the distributions calculated by the two methods were similar, that was, high variability occurred in densely populated areas with high AOD (EA and SA), and in arid areas affected by dust aerosols (Taklimakan desert, CA, and WA). Additionally, the temporal variability also captured the seasonal increase in AOD in SEA (mainly due to combustion), whereas this was almost not reflected in the spatial variability.

**Figure 12** showed the temporal and spatial relative AOD variabilities (standard deviation/average AOD) in different seasons to further study small-scale changes in aerosol. Compared with the AOD standard deviation, the relative variabilities of AOD can better show the degree of dispersion of data and be compared in regions with different AOD values. The high values of relative AOD variabilities generally appeared in areas affected by dust aerosol, such as WA, CA, SA, and northwest China. Among them, the highest AOD relative variability occurred in northwest China in the spring, and a long strip can be observed from the Taklimakan Desert in eastern China, suggesting a dust transport event under the action of the prevailing westerlies (Han et al. 2022). This phenomenon was also captured in the spatial relative AOD variability. During summer, an abnormally high value of relative AOD variability was found in NA, while the AOD value obtained from MODIS (**Figure 4**) did not increase significantly compared with other seasons in this region. The reason for this





phenomenon is that the summer wildfires in NA (contribution of ~17.33% in total Asia) are sporadic natural emission events, which hardly cause changes in long-term average seasonal AOD. However, the temporary AOD changes caused by wildfires

in NA were correctly captured by the relative AOD variabilities, which was also consistent with the higher OC and BC in the MERRA-2 data. In contrast, burning in spring and autumn in SEA did not lead to dramatic changes in relative AOD variability. This means that the aerosols in this region are still dominated by anthropogenic emissions and are relatively stable. The above finding shows that the abnormal increase in relative AOD variability has the potential to help find some natural emission events. It also implies when performing smoothing or constraining inversion of satellite aerosol parameters,

regions, which has different AOD deviations, should be given appropriate weights to reduce possible new errors while retaining features.

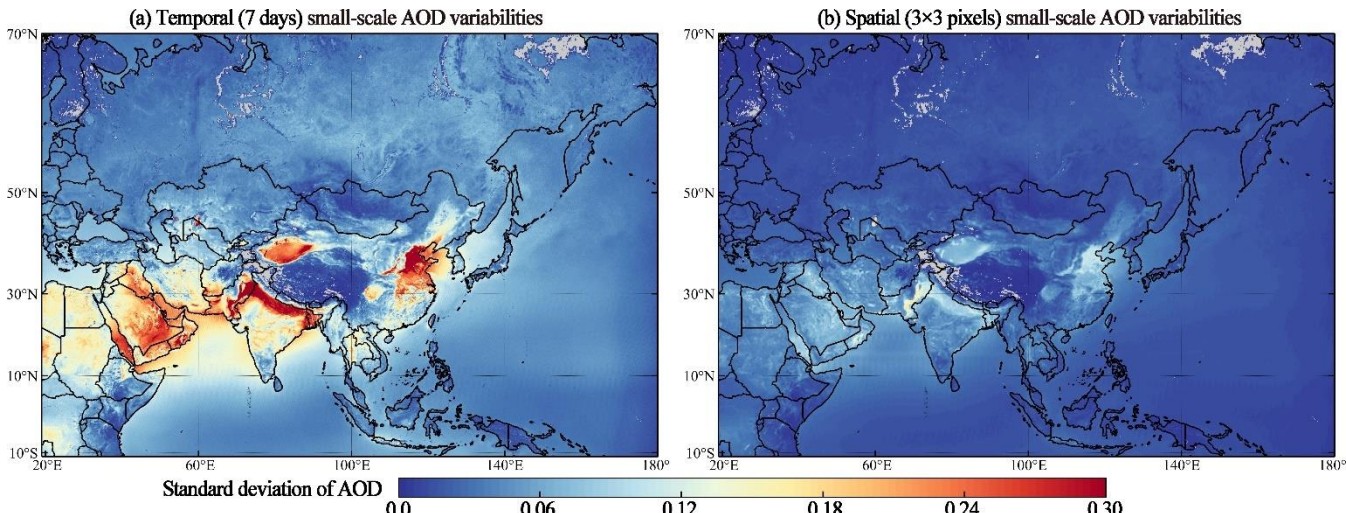

**Figure 11.** Temporal and spatial small-scale variabilities of AOD. The temporal standard deviation of AOD was calculated
in a range of 7 days, while the spatial standard deviation of AOD was calculated by a 3×3 pixel sliding window.

When we turn our attention to areas with high AOD values, especially China, it can be found that when the percentage of different size and type AOD is close, the relative AOD variability can observably vary. From the **Figure 12**, in all seasons, the relative AOD variability in NC centred on Beijing was always significantly higher than that in CWC and SC. Moreover,

two patches of high AOD values in the Sichuan Basin and Jianghan Plain could be clearly seen in the AOD distribution map, but these were not captured on the relative AOD variability distribution map. This is consistent with our previous finding (Zhang et al. 2021) that the air pollution events in Beijing are affected by significant external sources, such as northern dust transportation or the migration of pollutants from the south under abnormal climate. In contrast, cities in central China are in the centre of the pollution circle, and aerosols are less affected by circulation with a more stable distribution. Interestingly, a

similar phenomenon was observed in the plains of northern SA, where the value of AOD was high, but the relative AOD





variability was lower than that of southern SA and the EC, revealing a relatively stable aerosol environment. The relative AOD variability provides another perspective to analyse aerosol variation and improves our understanding of regional aerosol characteristics.

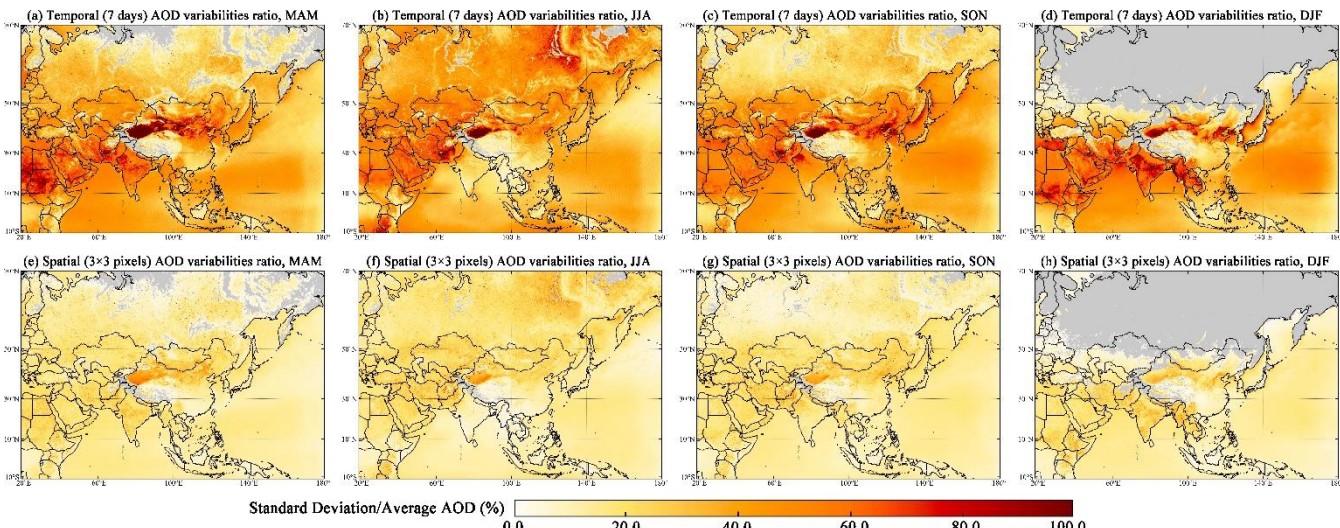


**Figure 12.** Temporal and spatial small-scale relative variabilities of AOD (standard deviation/average AOD). The temporal standard deviation of AOD was calculated in a range of 7 days, while the spatial standard deviation of AOD was calculated by a 3×3 pixel sliding window.

## 4 Summary and Conclusion

Distributions, long-term trends, and small-scale variabilities of aerosol particles with different size and type can reveal aerosol environmental characteristic in a region, which is also meaningful to improve the understanding of aerosols on the climate change, ecology, and public health. However, the attention to hot spots like Asia is not enough, and the perspective and depth of analysis can be further explored. In this study, multi-source aerosol products, including the ground-based observation (AERONET), satellite (MODIS and MISR), and re-analysis dataset (MERRA-2) were used for comprehensive

aerosol properties. The geometric mean method (Sayer and Knobelspiesse 2019), modified TFPW-MK test (Collaud Coen et al. 2020), and Pettitt change point test were applied to reappraise long-term trends and patterns of aerosols from a wide variety of angles. Additionally, we proposed a novel indicator for describing the small-scale variability of aerosols by calculating the standard deviation of AOD within a short period of time or space. This indicator can compensate for the suppression of extreme event weights by the geometric mean and complement the aerosol stability features that are often

ignored in the long time series analysis. In the meantime, the small-scale variability of aerosols also can act as an important prior constraint in advanced multi-pixel aerosol retrieval algorithms, which is usually ignored and not discussed in previous. The main conclusions of this study are as follows:





1) The MISR aerosol products perform best under clear air conditions (AOD < 0.25), but show increasing underestimation as AOD increases, especially in heavily polluted areas. This may be due to inappropriate aerosol models and an erroneous estimation of surface reflectance impacted by the reduced surface reflectance contrast at different angles and spectra at high AOD loading conditions. This means that the quantitative assessments of distributions and trends of SAOD, MAOD, LAOD are also likely to be underestimated.

2) Aerosols over land are dominated by fine particles in Asia. Specifically, the SAOD over land reaches its maximum in spring (0.094), followed by summer (0.076), and is lowest in winter (0.069) and autumn (0.059). The MAOD mostly distributes over the sea and it is very few over land with a mean value of ~0.008 and a small variation (±0.004) in a year. The annual mean value of LAOD is 0.033, showing a largely variation, depended on the source area of dust in Asia and the transmission during the monsoon season. The largest value of LAOD is found in SA (0.075) and the highest percentage of LAOD is in WA (37.6%).

3) Different types of aerosols are highly depended on regional environment and emission. The highest average value of sulphate AOD is found in SA (0.113) and followed by EA (0.086), showing association with a dense people population. Whereas, the peak value of sulphate AOD is in eastern China (> 0.4) of EA, where China's economy is growing rapidly. The dust aerosol mainly appears in dust source regions such as WA, CA, and northwest China. The most SS aerosol is distributed over seas, but one exception is that SS and dust aerosols will accumulate in the Arabian Sea with the summer monsoon, and invade the coastal areas around SA and WA. The OC and BC aerosols are usually produced by combustion and concentrated over EA, SA, and SEA. In addition, influenced by the wildfire events, high values of OC and BC AOD are also found in NA in summer, and this results in a ~10% increase in average OC AOD in Asian summer.

4) EA and SA emphasize two distinct contrasts in the analysis of long-term trends of aerosols over Asia. One is that the fastest growth of AOD appears in SA ($1.25\times10^{-3}$ per year), whereas the highest decrease in AOD is found in EA ($-5.28\times10^{-4}$ per year), though both regions are densely populated. Among them, the AOD growth in SA is mainly before 2009 while the AOD decrease in EA is around after 2013. Another is that an increase in carbonaceous aerosol is widely observed in Asia, with trends in OC and BC AOD of $2.00\times10^{-4}$ units/year and $2.20\times10^{-5}$ units/year, respectively. EA is an exception, where OC and BC AOD have decreasing trends, especially in central and eastern China.

5) In the last two decades from the perspective of trend, decreases in aerosol have been offsetting earlier increases in anthropogenic emission over Asia. From 2000 to 2020, the percentage of SAOD decreases by -3.34% and percentage of LAOD increases equivalently in Asia. From the perspective of the type-segregated AOD, the percentage of sulphate, dust, and SS AOD decreases by -3.07%, -5.51%, and -9.80%, respectively. And the decreases in sulphate and dust aerosols are mainly from NA and northern part of China, respectively. By contrast, the OC and BC increases with the percentage of 17.09% and 6.23%, respectively. The highest increase in OC was mainly from SEA in spring, NA in summer, and SA in winter, respectively, and the wildfire events in NA contribute ~17.33% of the total increase of OC AOD in summer.

6) The small-scale variabilities of AOD in temporal (7 days) are much higher than that in spatial (3×3 pixels), and the high values are mainly found in regions with high AOD such as EA and SA and dust source regions such as WA and Taklimakan

Desert. The increases in the relative AOD variabilities can imply natural aerosol emissions, such as dust transport events under the action of the prevailing westerlies in north China and anomalous wildfire events in NA, which are difficult to observe only from the seasonal distributions of AOD. In addition, in the same AOD high value areas, the relative AOD variability in NC is also significantly higher than that of CWC, SC, and northern SA. This means that the AOD in NC fluctuates strongly, in contrast, the aerosol changes in CWC, SC, and northern SA are more stable showing persistent air pollution.

This study provides a comprehensive insight into the distribution and trends of different sizes and types of aerosol loadings in regions of Asia, by combining data including ground-based stations, satellite observations and reanalysis data. A novel perspective of small-scale variabilities of AOD is proposed to study the changes of aerosol in a short period of time or space, which will reflect the degree of aerosol oscillation, imply natural aerosol emission events, and is expected to be used as a useful priory constraint for advanced multi-pixel retrieval algorithms. Therefore, the above findings greatly improve and complete understandings of aerosol environment and climatology in Asia.

**Data Availability**

The MODIS and MISR aerosol products were obtained from https://search.earthdata.nasa.gov/ (last access: 03 Nov 2022; Levy et al. (2013) and Garay et al. (2020)), the MERRA-2 aerosol products were obtained from https://disc.gsfc.nasa.gov/ (last access: 03 Nov 2022; Randles et al. (2017)), and the AERONET measurements were from https://aeronet.gsfc.nasa.gov/ (last access: 03 Nov 2022; Holben et al. (1998)). Data on Wuhan site can be provided by the corresponding authors upon request.

**Author Contributions**

SKJ and YYM designed the study, performed the analysis, and prepared the manuscript draft. BML, HL, WWY, and RNF collected the experimental data. ZWH and JPH reviewed the manuscript and given advices. WG and YYM supplied the site data and fundings.

**Competing Interests**

At least one of the (co-)authors is a member of the editorial board of Atmospheric Chemistry and Physics. The authors declare that they have no other conflict of interest.

**Acknowledgements**

The authors are grateful to the MISR, MODIS, MERRA-2 team, and AERONET principal investigators and site managers for kindly providing the valuable data.

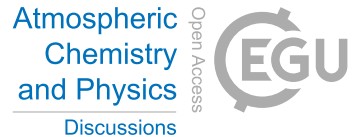

**Financial support**

This study was supported by the National Key Research and Development Program of China (grant no. 2018YFB0504500), the National Natural Science Foundation of China (grant nos. 42071348 and 42001291), and the Hubei Provincial Science and Technology Department Key Research Project (grant no. 2021BCA220).

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
