# Peer review of "A Comprehensive Reappraisal of Long-term Aerosol Characteristics, Trends, and Variability in Asia"

_Atmospheric Chemistry and Physics, 2023_

## Author Comment (AC1)

**Respond to Reviewer #2**

Dear reviewer, thank you for your useful comments. We have carefully studied your comments, and replied to your comments point by point and included corresponding modifications. In the following text, your comments are marked in bold italics, our responses are in black, and the modifications in manuscript are shown in blue.

**General comments**

**The manuscript by Jin et al. evaluated the distributions, trends, and variabilities of aerosol parameters (concentration, sizes, and types) using multi-source, long-term aerosol records, including ground-based observations, satellite products, and atmospheric reanalysis, which ensure the quality of the study and the conclusions. More importantly, geometric mean is used to better describe the lognormal distribution of aerosol, and a TFPW-MK method was introduced to avoid error in trend analysis of time-series data due to probable auto-correlation. Generally, the manuscript is well organized and written. However, before it could be considered to publish, there are still some problems that need clarification.**

Thanks for your comments and recognition. We have carefully revised our manuscript following your comments and we believe that our manuscript has been improved greatly.

**Specific comments**

**- Line 59: The statement is unclear for me. How could the short-term variances be beneficial to reveal the natural aerosol emissions? Please give a brief explanation.**

Thanks for your comment. We'd like to clarify that the short-term variability in aerosols we are discussing here are intended to reveal some events related to aerosol generation or transfer, and not necessarily actual study on aerosol emissions. Detecting aerosol emissions from satellite data is a complex and challenging issue that may require precise detection techniques and the use of model simulations [1]. Our motivation is to obtain pixel-level variability in the spatiotemporal distribution of aerosols through some simple calculations, which is probably meaningful about future studies on aerosol or particulate matter concentration estimation. For example, the GRASP algorithm we are studying allows for different spatiotemporal variabilities to be defined for each physical model parameter. From the results (Figures 12-13 in the manuscript), it appears that the short-term variability of aerosols we have studied is large in some known regions with significant aerosol emissions, such as the summer in northern Asia, dust source regions, and areas with severe urban pollution in China and India. More detailed study [2] has also given similar results, such as in urban polluted aerosol environments, where aerosol temporal variability is higher when wind speeds are lower due to reduced aerosol particle clearance by winds. So, our calculation of small-scale spatiotemporal variability of aerosols is primarily aimed at displaying a result which may be useful for estimating aerosol particle concentrations, rather than studying aerosol emission sources.

[1] Dubovik, O., Lapyonok, T., Kaufman, Y., Chin, M., Ginoux, P., Kahn, R., & Sinyuk, A.: Retrieving global aerosol sources from satellites using inverse modeling. Atmospheric Chemistry and Physics, 8, 209-250, doi:10.5194/acp-8-209-2008, 2008

[2] Chen, C., Dubovik, O., Schuster, G.L., Fuertes, D., Meijer, Y., Landgraf, J., Karol, Y., & Li, Z.: Characterization of temporal and spatial variability of aerosols from ground-based climatology: towards evaluation of satellite mission requirements. Journal of Quantitative Spectroscopy and Radiative Transfer, 268, 107627, doi:https://doi.org/10.1016/j.jqsrt.2021.107627, 2021

**- Line 224: What is the type I error, and please give a brief explanation.**

Thanks for your comment. The Type I error means that the null hypothesis is rejected when it is actually true, which results in a false positive result. In this study, Type I errors are mainly caused by the spatiotemporal autocorrelation of the data. The increase in Type I errors means that we obtain false significant trends of aerosol, which may lead to incorrect conclusions. Therefore, we used various methods (TFPW-MK and FDR) to avoid increasing Type I errors and obtain reliable conclusions.

To facilitate readers' understanding, we made the following modifications:

(Lines 225-228) However, the spatiotemporal auto-correlation is usually widespread in observed AOD series (namely, the data are not strictly independent and random). This makes the original MK test usually results the occurrences of type I error (rejecting true null hypothesis) in statistics and a large number of fake significances (Von Storch 1999).

**- Line 296: This study used the AOD geometric to investigate the temporal trends, which is an essential point for the work. Meanwhile, it appears they use normal mean for standard deviation calculations. Does this lead to any biases?**

Thanks for your comment. Yes, using geometric mean to process data is a feature of our study, but it will not be contradictory to using standard deviation to evaluate small-scale variability of aerosols. Because the data organization method of geometric mean is mainly for studying the spatial distribution and trend of aerosols, which is not a same topic comparing to the small-scale variability. More importantly, the standard deviation is a simple statistical measure that is not related to the data distribution. This means that when we only want to calculate the standard deviation without any additional hypothesis, the distribution of the data does not need to be considered.

**- Section 4.1 is helpful to understand the quality of different products before looking at the long-term trends. However, some further analysis/discussion would be beneficial for the readers. For example, how much data is removed due to cloud cover and/or unfavorable land cover such as snow? How does this impact the analysis of the long-term trends? Does this lead to more or less data in some areas during the same season?**

Thanks for your comments. Yes, cloud and snow cover or missing data can have an impact on statistical results. For MERRA-2, all data are continuous and complete; for satellite data, the cloud coverage rate is around 60%, and snow usually impacts aerosol retrieval in high latitude area during winter. But these issues do not affect the main results because we have taken into account the seasonal variations in both spatial distribution and trend analysis (using seasons or months is also recommended compared to analyzing on an annual basis [1]). In addition, in the long-term trend analysis based on rank, data missingness is also acceptable due to the advantages of the TFPW-MK method [2].

To avoid unnecessary misunderstandings, we have added some explanations at the relevant locations. The specific modifications are as follows:

(Lines 395-399) Here, it is noted that due to snow cover, there was a large number of missing retrievals in winter of NA, which would lead to an overestimated mean AOD for the whole Asia. Therefore, the specific details of seasonal variations in aerosol should be obtained from the distribution maps of AOD. Moreover, since the rank-based trend analysis method allows for missing data, the missing values caused by cloud cover and snow cover do not have a significant impact on the calculations of trends.

[1] Collaud Coen, M., Andrews, E., Bigi, A., Martucci, G., Romanens, G., Vogt, F.P.A., & Vuilleumier, L.: Effects of the prewhitening method, the time granularity, and the time segmentation on the Mann–Kendall trend detection and the associated Sen's slope. Atmospheric Measurement Techniques, 13, 6945-6964, doi:10.5194/amt-13-6945-2020, 2020

[2] Yue, S., Pilon, P., Phinney, B., & Cavadias, G.: The influence of autocorrelation on the ability to detect trend in hydrological series. Hydrological Processes, 16, 1807-1829, doi:10.1002/hyp.1095, 2002

**- The AOD was higher in spring than winter due to the emissions associated to heating and industry and lower boundary layer? How much of this is due to cloud cover vs. dust from deserts?**

Thanks for your comments. There is a misunderstanding here. It has been shown that industrial emissions and lower boundary layers can exacerbate regional air pollution events, but these issues do not apply to studies of total AOD (total atmospheric column) in Asia. We just want to briefly remind the readers that, due to the missing satellite data, the average total AOD in winter may be biased towards higher values because of the missing data from North Asia (where anthropogenic emissions are relatively low). However, it is actually very difficult to assess the impact of data gaps or dust on total AOD at such a large scale. If we look at it from the perspective of MERRA-2 (where the data is continuous and complete): the average AOD in Asia is highest in spring (0.215) and follow by summer (0.200), autumn (0.140), and winter (0.133); the average dust AOD in Asia is highest in spring (0.059), followed by summer (0.036), autumn (0.023), and winter (0.022).

**- As another note on the aerosol type, the authors report decreasing or increasing dust and sea salt. But why these are decreasing or increasing as the emissions of these two aerosol types are natural and not anthropogenic.**

Thanks for your comments. Yes, in general, natural sources of aerosols are rarely influenced by human activities. However, some human factors may alter the distribution of aerosols, with the most typical example being Beijing's protective policies against the northwestern transport of dust in the spring. As shown in Figure 10 in manuscript, from a perspective of long-term trend analysis, this has led to a decrease in dust aerosols in the Beijing region.

Additionally, in the previous version, we overlooked the spatial autocorrelation of the data, which could lead to false significant trends. After applying the FDR method, many false trends related to dust and sea salt have been eliminated.

**- Line 105-109: Except for the consideration of adjacent pixels, a globally consistent assumption is also very important for continuous distribution in aerosol retrieval.**

Thanks for your comment. Yes, in retrieval, the assumption of global consistency is important for the continuous distribution of aerosols, but it is challenging to achieve.

**- Section 2.2: Please clarify the estimated uncertainties of different parameters from observation of CE-318.**

Thanks for your comment. we have revised the manuscript and added a description of the uncertainties:

(Lines 133-143) The version 3 AERONET aerosol products did a lot of improvements based on the version 2, such as applying a new polarized radiative transfer code, updating datasets of spectral solar flux and surface albedo parameters, modifying temperature and gas absorption corrections, and using stricter quality control rules (Giles et al. 2019; Sinyuk et al. 2020); The bias of AOD estimation is +0.02 and the uncertainty with one standard deviation is ~0.02; The uncertainty of SSA is estimated ~0.03 when the AOD at 440 nm is larger than 0.4. In addition, an extra site established at Wuhan University was also applied in this study (Jin et al. 2021) to supplement the aerosol optical properties in CC area. This site is equipped with the same sun sky photometer as AERONET and calibrated annually using the China Meteorological Administration Aerosol Remote Sensing Network (CARSNET) (Che et al. 2009). The AOD was calculated from the direct sun measurement and other complex properties were retrieved from the sky irradiance under cloudless conditions (Smirnov et al. 2000), with the method of Dubovik and King (2000). The criterion of data is similar to the Level 1.5 Version 2.0 product released by the AERONET.

**- How different it is if using a normal distribution in the long-term aerosol studies compared with using a lognormal distribution.**

Thanks for your comment. It is difficult to use a unified standard to evaluate the differences between geometric and arithmetic means, and these differences are mainly concentrated in areas with high aerosol loadings. Following report by Sayer [1], the differences between geometric and arithmetic means of aerosol (**Fig. R1**) can change a lot depending on the time scale of data aggregation and the magnitude of AOD. For daily data, the difference is from 0 to ~0.02; for monthly data, the difference can be magnified by orders of magnitude.

[Figure]

**Fig. R1**. Difference between geometric and arithmetic means of AOD (Figure 9 in Sayer et al. (2019) [1])

[1] Sayer, A.M., & Knobelspiesse, K.D.: How should we aggregate data? Methods accounting for the numerical distributions, with an assessment of aerosol optical depth. Atmospheric Chemistry and Physics, 19, 15023-15048, doi:10.5194/acp-19-15023-2019, 2019

**- Line 280: What do the 'pixel level' and 'region level' refer to? Please clarify.**

Thanks for your comments. The 'pixel level' means that trends are estimated based on the grids (such as Figure 9-10 in the manuscript), and 'region level' means that trends are estimated based on sub-regions in Asia, such as EA, SEA, and SA et al.

**- Figure 10: Is the aggregation method of aerosol percent also the geometric?**

Thanks for your comments. Yes, as shown in Eq. 16 in the manuscript, the mean values and trends are all calculated based on the geometric mean method.

**- Table 2: The SS aerosol in SEA showed a slight decrease but this was not revealed in size-segregated AOD.**

Thanks for your comments. We want to clarify that in the previous version of our study, the lack of consideration of spatial-correlation in the data may have led to some false trends. With the addition of the FDR method, the trends obtained are more accurate in this version of the manuscript. Therefore, these false trends shown by the SS aerosol have been largely eliminated in this version. Additionally, the MERRA-2 data is more continuous in both time and space, resulting in more detected trends. However, the SS aerosol from MERRA-2 and size-segregated aerosols from MISR are not one-to-one correspondence, so the analysis shows as different a result. Therefore, we used both satellite and reanalysis data to obtain a comprehensive estimation of aerosols.

---

## Author Comment (AC2)

**Respond to Reviewer #1**

Dear reviewer, thank you for your useful comments. We have carefully studied your comments, and replied to your comments point by point and included corresponding modifications. In the following text, your comments are marked in bold italics, our responses are in black, and the modifications in manuscript are shown in blue.

**General comments and recommendation**

**This paper uses MODIS, MISR, and MERRA2 aerosol data to examine aerosol trends across Asia over the period 2000-2020 (plus AERONET to evaluate the satellite retrievals, and MODIS fires for additional context). There are a lot of papers on aerosol trends published, particularly covering Asia owing to its high population and complicated aerosol system which continues to evolve. This means it is important to question whether a submission really brings anything new to the state of knowledge. This study identifies itself as a "comprehensive reappraisal" and I believe in this case it is warranted. While these data sets are often used for trend analyses, this study approaches trend calculations using different statistical approaches than previous studies (specifically: analyses based on geometric rather than arithmetic means; autocorrelation-resistant trend and significance estimates). This compensates for some of the quirks of aerosol data (i.e. skewed distributions and autocorrelated data) which are often neglected in similar studies, which is a strength. It also accounts for the possibility in change points in trends, which is reasonable as with a 20-year time series it is reasonable to expect that trends might not continue linearly the whole time. The results and previous studies are discussed in the context of the methodological differences.**

Thanks for your recognition. It is no doubt that your comments and suggestions are valuable, and we can learn a lot from them. We have carefully revised our study in accordance with your comments, which really helps us improve the quality of our manuscripts, especially in the problem of the false discovery rate. Specific revisions and responses are shown in the following text.

**The topic is relevant and appropriate for the journal. The quality of written language is ok (standard journal copy-editing should be sufficient). The quality of the figures is in general ok. The data availability statement is present and mostly sufficient (I am not sure that "available on request for the Wuhan data is acceptable – I believe Copernicus publications now requires all data are on a public repository unless there is a compelling reason otherwise). The supplementary materials are relevant.**

Thanks for your recognition and comment. We promise to abide by the publication policy of Copernicus; if necessary, we are willing to publish the data on the Wuhan site. To be honest, the Wuhan site in this study is only used as a case on behalf of gourd-based observation in Central China. So, we only display it in the supplementary materials. Details of the Wuhan site are shown in the response of comment 3.

**That said, there is some missing information needed to understand the study, and I have some typographical and figure suggestions, and a few questions. Some of this could affect the conclusions, so I favour major revisions and re-review. I would be happy to provide a further review. I note my expertise is on the satellite**

**retrieval and analysis side and not on the aerosol emissions/transport/policy side, so I recommend at least one other reviewer is an expert in those domains in case I have missed something in my review.**

Thanks for your reviewing. We have made revisions based on your comments carefully. We believe our manuscript has been greatly improved.

**Specific comments**

1. **Lines 105-115: I am not sure how relevant these lines are in the context of the study. It is true that algorithms like GRASP and MAIAC apply spatial and/or temporal smoothness constraints. In principle this could artificially decrease aerosol variability, but in practice I don't think this is a major concern, as many aerosol events extent for more than one pixel in space or time. From my use of these data sets, such constraints instead work to reduce noise and to reduce artefacts resulting from e.g. sporadic cloud contamination. So that would be a net benefit to trend analysis as there would be fewer artificial (positive) outliers – the current wording of the paper implies that these smoothness constraints are a problem. I suggest this text is modified or removed.**

Thanks for your comments. Yes, we agreed with you that these sentences were not closely related to the focus of our research, and we cannot infer or imply any views without sufficient evidences. Therefore, we have removed these sentences following your recommendation.

In fact, our realization that "spatial and/or temporal smoothing constraints" can influence on aerosol retrieval results is inspired by our previous practice and experiment with the GRASP algorithm. When running the GRASP algorithm, these 'spatial and/or temporal constraints' can be customized in the configuration file by modifying the values and derivative orders of Lagrange multipliers. In the ideal implementation of GRASP, it is expected to use a same configuration file (this means that the 'spatial and/or temporal constraints' are also same) for the global aerosol retrievals, because the GRASP emphasizes to use a uniform aerosol/surface model assumption (independent with regions or seasons) in order to obtain more continuous global distributions (especially for AOD around the coastline) of aerosol parameters. However, based on our study experience and consensus in Asia, the aerosol can change greatly. Therefore, we expect to give a more detail constraint in aerosol retrieval by studying the small-scale variability of aerosols, and obtain better results in some regional studies. This is the reason that we think a global uniform constraint on AOD change will influence the aerosol retrieved. Fortunately, with your reminding, we are aware that this opinion may be arbitrary and is not closely related to the topic of our study, and so we have deleted them.

2. **Section 2.2: the AERONET references here are outdated. For the current version 3 direct sun (AOD) data set is Giles (2019): https://amt.copernicus.org/articles/12/169/2019/ The inversion data set (size, SSA/AAOD) is Sinyuk et al (2020): https://amt.copernicus.org/articles/13/3375/2020/**

Thank you very much for your comments and suggestions. We learned a lot from these two references and updated them in the manuscript following your comments. We believe this revision will also be beneficial for our readers.

The specific modifications are shown as following:

(Lines 133-138) The version 3 AERONET aerosol products did a lot of improvements based on the version 2, such as applying a new polarized radiative transfer code, updating datasets of spectral solar flux and surface albedo parameters, modifying temperature and gas absorption corrections, and using stricter quality control rules (Giles et

al. 2019; Sinyuk et al. 2020); The bias of AOD estimation is +0.02 and the uncertainty with one standard deviation is ~0.02; The uncertainty of SSA is estimated ~0.03 when the AOD at 440 nm is larger than 0.4.

3. **Lines 144-145: Could the authors provide more information on the Wuhan site? It does not look like it is part of the official AERONET (checking on their website). The paper says it is the same instrument type – what about data processing? Does it run on the AERONET processing code or something else? How is it calibrated? The Jin paper cited provides a calibration reference (which should be included here too); it says it uses the same algorithms as AERONET but it's still not clear whether that means the same code or a different implementation of the same approach.**

Thank you very much for your comments. Yes, this site was set up on the roof of our laboratory (114°21′E, 30°32′N) by our team in Wuhan University, and it was not connected to the official AERONET network due to the local policy of government. But unofficial reporting or sharing of aerosol observation results from this site for research purposes is not prohibited. Therefore, we did some studies about aerosol based on the Wuhan site, such as Ma et al. (2019) [1].

In the following, we will provide all information about this site as much as we know:

1) The location of our site is in Wuhan University (114°21′E, 30°32′N). It is on the roof of the State Key Laboratory of Information Engineering in Surveying, Mapping and Remote Sensing (LISRMARS), with the height above sea level of ~ 52 m (**Fig. R1**). Because the site location is in the city centre, the aerosol source is mainly from anthropogenic emissions, such as automobile exhaust and process of building construction.

[Figure]

**Fig. R1.** Location of Wuhan University and our CIMEL sun sky photometer site from Google Earth.

2) The instrument at our site is the CIMEL Sun Sky photometer, the same as AERONET. The maintenance is mainly based on the China Meteorological Administration Aerosol Remote Sensing Network (CARSNET) [2], which is an official platform affiliated with the China Meteorological Administration. In general, we will send our photometer to the CARSNET for calibration every year. The calibration can be mainly divided into two steps: 1) indoor correction by the standard laboratory integrating sphere; 2) joint observation with standard CIMEL sun photometers which are installed in China Meteorological Administration. These standard CIMEL sun photometers were calibrated using the Photome trie pour le Traitement Ope rationnel de Normalization Satellitaire (PHOTONS) calibration facilities in Lille (France), and Carpentras (METEOFRANCE). And the PHOTONS master instrument was calibrated by Langley plot analysis at Izana Observatory (AEMET, Spain) following the calibration protocol used by NASA staff. Therefore, the measurement from CARSNET is reliable, and it has great consistency with AERONET.

3) The AOD data is calculated from the Sun Triplet Measurements (NSU) with the ASTPwin software. The complex aerosol properties, such as SSA, are retrieved from the Right/Left Almucantar Sky Irradiances (ALR/ALL), with the method of Dubovik and King 2000 [3], and the specific software package for algorithm implementation is also from the CARSNET [4]. Strictly speaking, the results from our site are equivalent to the AERONET Version 2.0 and Level

1.5 products. Therefore, theoretically, as reported by the previous studies [5-6], the uncertainty of AOD is 0.01 to 0.02, and the uncertainty of SSA is 0.03 to 0.07 depending on aerosol loading and absorbing.

Following your comments, we have added more details about our sites in the manuscript. Due to the ground-based data is also not the main point in our study, please allow us to explain only what is necessary in the manuscript:

(Lines 138-144) In addition, an extra site established at Wuhan University was also applied in this study (Jin et al. 2021) to supplement the aerosol optical properties in CC area. This site is equipped with the same sun sky photometer as AERONET and calibrated annually using the China Meteorological Administration Aerosol Remote Sensing Network (CARSNET) (Che et al. 2009). The AOD was calculated from the direct sun measurement and other complex properties were retrieved from the sky irradiance under cloudless conditions (Smirnov et al. 2000), with the method of Dubovik and King (2000). The criterion of data is similar to the Level 1.5 Version 2.0 product released by the AERONET.

[1] Ma, Y., Zhang, M., Jin, S., Gong, W., Chen, N., Chen, Z., Jin, Y., & Shi, Y.: Long-Term Investigation of Aerosol Optical and Radiative Characteristics in a Typical Megacity of Central China During Winter Haze Periods. Journal of Geophysical Research: Atmospheres, 124, 12093-12106, doi:10.1029/2019jd030840, 2019

[2] Che, H.Z., Zhang, X.Y., Chen, H.B., Damiri, B., Goloub, P., Li, Z.Q., Zhang, X.C., Wei, Y., Zhou, H.G., Dong, F., et al. Instrument calibration and aerosol optical depth validation of the china aerosol remote sensing network. J. Geophys. Res.-Atmos., 2009, 114, 12, doi:10.1029/2008JD011030.

[3] Dubovik, O., & King, M.D.: A flexible inversion algorithm for retrieval of aerosol optical properties from Sun and sky radiance measurements. Journal of Geophysical Research Atmospheres, 105, 20673-20696, doi:10.1029/2000JD900282, 2000

[4] Che, H.Z., Qi, B., Zhao, H.J., Xia, X.G., Eck, T.F., Goloub, P., Dubovik, O., Estelles, V., Cuevas-Agullo, E., Blarel, L., Wu, Y.F., Zhu, J., Du, R.G., Wang, Y.Q., Wang, H., Gui, K., Yu, J., Zheng, Y., Sun, T.Z., Chen, Q.L., Shi, G.Y., & Zhang, X.Y.: Aerosol optical properties and direct radiative forcing based on measurements from the China Aerosol Remote Sensing Network (CARSNET) in eastern China. Atmospheric Chemistry and Physics, 18, 405-425, doi:10.5194/acp-18-405-2018, 2018

[5] Eck, T.F., Holben, B.N., Reid, J.S., Dubovik, O., Smirnov, A., O'Neill, N.T., Slutsker, I., & Kinne, S.: Wavelength dependence of the optical depth of biomass burning, urban, and desert dust aerosols. Journal of Geophysical Research: Atmospheres, 104, 31333-31349, doi:10.1029/1999jd900923, 1999

[6] Dubovik, O., Smirnov, A., Holben, B.N., King, M.D., Kaufman, Y.J., Eck, T.F., & Slutsker, I.: Accuracy assessments of aerosol optical properties retrieved from Aerosol Robotic Network (AERONET) Sun and sky radiance measurements. Journal of Geophysical Research: Atmospheres, 105, 9791-9806, doi:10.1029/2000jd900040, 2000

4. **Section 2.3: the spatial resolution of the satellite data sets is described here, but what is not clear is how they are aggregated in space-time for the trend analyses later. It can't be a simple stacking because the products are on swath-referenced grids, and the trend analysis would need reprojection to an Earth-referenced grid. This is also important for the later analyses of spatiotemporal mapping. It is a simple nearest-neighbor remapping or is there averaging if multiple pixels fall within a grid element on a given day? What is the spatial size of the grid – 10 km equal area or 0.1 degree equal angle, for example? Does it vary dependent on data set? How are MODIS Terra and Aqua data combined, are they treated as one data set? This should all be stated in the manuscript.**

Thank you very much for your comments and reminding. Yes, the data aggregation in our study followed the method of sayer et al (2019) [1] that aggregating first in space and then in time. As sayer et al (2019) points out that this way is similar to the way polar-orbiting L3 aggregates sample the global aerosol system (as each L2 product is essentially a near-instantaneous snapshot).

Specifically, for the MISR and MODIS data, the AOD is remapped from their native spatial resolution for 0.1° (equal angle), with the nearest-neighbour method. Here, the Terra and Aqua MODIS products are treated as one data set and the aggregation method dose not vary depending on the data set. And then, the AOD data in each pixel is aggregated into monthly data by time scale with the geometric mean method (without aggregating into daily data before). Therefore, there is not an averaging process if multiple pixels fall within a grid element in a given day, and all the trend analysis is done based on the monthly data sets.

We are sorry that this part was omitted in the manuscript. The specific revision is shown below:

(Lines 210-213) Specifically, the AOD data from MODIS and MISR will firstly be remapped from their native spatial resolution for 0.1° using the nearest-neighbour method. Here, the Terra and Aqua MODIS products are treated as one data set. Then, the AOD data in each pixel is directly aggregated into monthly data by time scale with the geometric mean method, and all the trend analysis presented below are done based on these monthly data sets.

[1] Sayer, A.M., & Knobelspiesse, K.D.: How should we aggregate data? Methods accounting for the numerical distributions, with an assessment of aerosol optical depth. Atmospheric Chemistry and Physics, 19, 15023-15048, doi:10.5194/acp-19-15023-2019, 2019

5. **Line 152: the reference for Collection 6.1 MODIS Deep Blue is Hsu et al (2019): https://agupubs.onlinelibrary.wiley.com/doi/full/10.1029/2018JD029688 the paper is about VIIRS and MODIS. The 2013 paper cited is about collection 6.**

Thanks for your comments and reminding. We have updated the new reference and revised this mistake:

(Lines 148-150) Two operational aerosol retrieval algorithms: Dark Target (DT) and Deep Blue (DB) have been applied to produce high-accuracy C6.1 Level 2 aerosol products, with the spatial resolutions of 3 km and 10 km (Hsu et al. 2019; Levy et al. 2013).

6. **Line 190: the text about GCOS doesn't really make sense as written. I know what the authors mean but the wording needs improvement. This uncertainty represents a goal uncertainty for an aerosol climate record. GCOS itself is an organization, not a metric. Words like "goal uncertainty" or similar should be added into that sentence. I also suggest not naming the metric "GCOS" maybe something like %GCOS (with GCOS subscripted) would make it clearer that this is a percentage metric relating to GCOS goals.**

Thanks for your advice about the writing. That we used the goal uncertainty presented by GCOS was to better evaluate different aerosol products. This is a little bit similar to using the Expect Error (EE%), which presents the error envelop under the assumption of normal distribution within one sigma confidence interval, in the aerosol retrieval related studies. To avoid potential misunderstandings and make the text more meaningful, we have revised these sentences following your advice. The specific revision is below:

(Lines 188-190) In addition, the percentage (%GCOS) falling into the goal uncertainty presented by Global Climate Observing System (Popp et al. 2016) is also considered to better compare different AOD data in the long-term and large-scale analysis. For AOD data from satellites, the goal uncertainty is defined as 0.04 (assuming an uncertainty in any AERONET measurement of 0.01) or 10% depending on which one is larger (Eq. 4). Finally, the Spearman's rank correlation coefficients ($\rho$) and the Pearson's correlation coefficients (R) were also used to assess relationship between AERONET and satellite measurements with a significance (p) test to confirm whether the satellite retrieval result was available.

**7.** **Section 3.1 and 4.1: linear least squares regression is not appropriate for this type of analysis because e.g. the data distribution is skewed, the errors are dependent on AOD magnitude (as seen in Figure 2), there can be multiple different sub-populations with different error characteristics at a single site or region, the data are autocorrelated, etc. All of these violate the assumptions needed for the least squares slope and offset estimators to be unbiased. Calculating Pearson's linear correlation coefficient is ok (I personally prefer Spearman's rank correlation) but the slope and offset calculated are likely biased and misleading and so should be removed from the manuscript otherwise it is not statistically sound. I don't believe they are necessary for the argument about uncertainty any bias anyway, as relevant information is provided by the other statistics calculated and by the histograms in Figure 2. I have strong opinions on this point because inappropriate statistics are too common in papers and is sometimes justified by saying that other papers did it. The discussions of regression slope and intercept should be removed from the paper.**

Thank you very much for your comments and reminding. Yes, we are already aware that the slope and intercept of linear regression are inappropriate here, because the distribution of the data indeed can have a big effect on the slope. It is very common to use linear regression to evaluate AOD data in previous studies, so that we missed the probability problems in initial. We also agree with you that using the Spearman's rank correlation is better for the trend analysis is also based on the rank. Therefore, we reorganized this section, deleting the discussion about the slope and intercept of linear regression and calculating the Spearman's rank correlation. The specific revision is shown below:

[revised manuscript text omitted]

8. **Section 3.2.2.: I have read this text and the accompanying supplementary figures a few times and am still not sure I understand it. I recommend rewording – maybe more detail is needed in the supplement to explain these figures, and some of the explanatory text could come after the equations instead of before. Moving Figure S2 to the main paper would also be useful. I am a bit lost on how the p-value is calculated in that figure. If I understand correctly the threshold represents the magnitude of autocorrelation you would be no more than 5% likely to observe if there were a true zero underlying autocorrelation? So the distributions narrowing and centering around zero mean that the method goes from having about 20% of grid cells with correlation smaller than this threshold, to about 90%? Also, Equation 10 implies that**

**correcting for (positive) autocorrelation makes the corrected trend estimate (beta prime) bigger than the initial trend estimate (beta) – i.e., unaccounted for autocorrelation would have made the trend appear smaller than it really is. Is that correct or is that backwards?**

Thanks for your comments and questions. We will answer your questions first and then make revisions according to your comments. 1) Yes, your understanding is correct. The significance of autocorrelation is in accordance with the general explanation of significance. If a set of data falls into this two-tailed range, the probability that it has the serial autocorrelation is less than 5%; Or, in all the data that fall into this two-tailed range, only < 5% percent of them will show the serial autocorrelation. The estimation method of autocorrelation significance (p-value in Figure S2) is from Anderson in 1942 [1]; the equations are shown below (Eq. R1 for p < 0.05 and Eq. R2 for p < 0.01).

$$\frac{-1 \pm 1.645\sqrt{(N-2)}}{N-1} \quad \text{(R1)}$$

$$\frac{-1 \pm 2.326\sqrt{(N-2)}}{N-1} \quad \text{(R2)}$$

The two-tailed range of autocorrelation significance only depends on the length of the series (N).

2) Yes, from the Figure S2, after the Pre-Whitening (or Trend-Free Pre-Whitening), the percent of grid cells falling into the threshold [-0.1087,0.1006] of autocorrelation significance (p < 0.05) increases from ~20% to ~90%. This means that the influence of serial autocorrelation is greatly reduced. We can see this clearly from Yue et al (2002)'s results [2], as Fig. **R2**.

[Figure]

**Fig. R2**. Effect of positive serial correlation on the type I error: (a) unfiltered; (b) pre-whitened (Figure 1 in Yue et al. (2002) [2])

These results are calculated by the Monte Carlo method. It can be found that for original data (Fig. R2a), the type I error (p < 0.05) is ~5% for Lag-1 serial correlation coefficient = 0 and it increases to 50-60% when the Lag-1 serial correlation coefficient = 0.8. The type I error means rejecting the true null hypothesis (rejecting the true no trend hypothesis; namely, data without a trend is considered to have a trend), and thus the existing of type I error will cause any data series without significant trends to look trend-significant (overestimating trend significance). After the Pre-Whitening process (Fig. R2b), the type I error is suppressed largely.

3) First of all, we must apologize because we left out a division sign when we were editing the formula. This may be the reason to make you feel confusing. The revised formula is shown as below (Eq. R3).

$$\beta' = \beta / \sqrt{(1 + r_1^D)/(1 - r_1^D)} \quad \text{(R3)}$$

We have checked our code and fortunately it is typed correctly in the code. Back to your question, in fact, unaccounted for the series autocorrelation would have made the trend appear larger than it really is. More specifically, this 'larger' can be divided into two parts. One is the type I error discussed above which can probably cause some data series without significant trends to look trend-significant (overestimating trend significance). The other is the original Sen's slope, namely $\beta$. Wang et al (2015) [3] suggests that the Sen's slope method will result a larger value of trend when processing a serial autocorrelation data (overestimating trend magnitude) and he recommends to correct the original Sen's slope by the Lag-1 serial correlation coefficient (Eq. R3). Collaud Coen et al (2020) also confirms this, and she finds that the correction of Sen's slope is also meaningful in the atmosphere related data. Therefore, these processing and discussion about the series autocorrelation in our manuscript are designed to minimize potential overestimation of the trend as much as possible and give a 'conservative' result.

Finally, following your comments, we moved the Figure S2 to the main text and gave additional specific explanations to these figures both in the main text and supplementary text. We also reorganized the section 3.2.2 to make it easier for readers to understand. The specific revisions are shown below:

(Lines 225-228) However, the spatiotemporal auto-correlation is usually widespread in observed AOD series (namely, the data are not strictly independent and random). This makes the original MK test usually results the occurrences of type I error (rejecting true null hypothesis) in statistics and a large number of fake significances (Von Storch 1999).

(Lines 234-235) Taking the monthly MODIS AOD as an example, **Figure 2** shows the serial auto-correlation using different data processing methods.

(Lines 237-238) Therefore, we can use this lag-1 serial coefficient to evaluate the auto-correlation of a time sequence of aerosols.

(Lines 245-247) The $t$ represents the time position of data in the series and the $T$ is the length of data (namely, there is $t$ = 1, 2, 3, …, $T$). Therefore, we can use this lag-1 serial coefficient to evaluate the auto-correlation of a time sequence of aerosols.

(Lines 248-250) **Step 2**. If the $r_1^o$ fall into the significant intervals (p < 0.05), the $X_{t,T}$ is considered to be serial auto-correlated. Then, before estimating the significance using the MK test, it is need to calculate the Sen's slope ($\beta$), remove the serial trend, and create the blended series (namely the processing of TFPW), as Eq. 7-9.

(Lines 256-257) This process can be understood simply as removing the auto-correlation part from the time series data. Then, the significance of trends and abrupt change points are assessed from the blended series.

(Lines 260-264) **Step 3**. Finally, the variance corrected trends ($\beta'$) can be approximately estimated by $\beta$, as Eq. 10.

$$\beta' = \beta/\sqrt{(1+r_1^D)/(1-r_1^D)} \tag{10}$$

In this study, the β^' is used to present the AOD trends with different size and type aerosols. This method can be considered to keep a good balance between maintaining a low type I error and a relatively strong power of trend detection, which also provides a robust choice in studies of trend analysis (Wang et al. 2015).

[1] Anderson RL. 1942. Distribution of the serial correlation coefficients. Annals of Mathematical Statistics 13(1), doi: 10.1214/aoms/1177731638

[2] Yue, S., Pilon, P., Phinney, B., & Cavadias, G.: The influence of autocorrelation on the ability to detect trend in hydrological series. Hydrological Processes, 16, 1807-1829, doi:10.1002/hyp.1095, 2002

[3] Wang, W., Chen, Y., Becker, S., & Liu, B.: Variance Correction Prewhitening Method for Trend Detection in Autocorrelated Data. Journal of Hydrologic Engineering, 20, 04015033, doi:10.1061/(ASCE)HE.1943-5584.0001234, 2015

[4] Collaud Coen, M., Andrews, E., Bigi, A., Martucci, G., Romanens, G., Vogt, F.P.A., & Vuilleumier, L.: Effects of the prewhitening method, the time granularity, and the time segmentation on the Mann–Kendall trend detection and the associated Sen's slope. Atmospheric Measurement Techniques, 13, 6945-6964, doi:10.5194/amt-13-6945-2020, 2020

9. **Tables: in general I think too many significant figures are given in these, it adds clutter and implies more precision on the metrics (these are all estimates of the population true metric) than we likely have given sample sizes. For example in Table 1 do we really need the GCOS goal percentage to 5 significant figures? I suggest two decimal places for MAE, RMB, RMSE, and R, and rounding to the nearest 1% for the GCOS goal.**

Thanks for your comments. We have revised the manuscript following your suggestions. The specific revisions have been shown in the response to question 7.

10. **Lines 327-328: The wording about a significance test is a little strange. Is this saying that because AERONET and MISR AAOD are essentially uncorrelated (R=-0.023) it's not used in the trend analysis because it can't capture the spatiotemporal variation? If so, I would say it that way. Since AERNOET AAOD is also a retrieved (not measured) parameter, and has significant uncertainty, it would be worth stating that some of this may be due to limitations on the AERONET side and not just the MISR side.**

Thanks for your comments and queries. Yes, we understand that the AAOD from AERONET also has relatively large uncertainty (depending on AOD and SSA), and we also understand that it is very challenging to retrieve AAOD from satellite observation. But in general, evaluating satellite aerosol products by comparing to the AERONET data is one of the most recognized practices. Here, we don't want to say that MISR's AAOD is no good and we only exclude this parameter from the study only out of an abundance of caution. In fact, the Spearman's rank correlation coefficient of AAOD is ~0.1 with p = 0.13 (close to 0.1, 90% confidence interval). This illustrates the potential of this parameter.

To avoid misleading our readers, we have added an additional explanation in accordance with your comments:

(Lines 360-364) In contrast, the AAOD had the lowest correlation coefficients (R = -0.02 and $\rho$ = 0.10), and they also cannot pass the significance test. Therefore, the AAOD was removed from this study. It should be clarified that this does not mean that AAOD cannot correctly reveals spatial variation of absorbing aerosols, because the uncertainty can also partially come from the retrievals of AERONET. But out of an abundance of caution, the removing of AAOD data in this study is to avoid introducing uncertainty beyond the expectation and to make the results more interpretable in trend analysis.

11. **Figure 7: I like this figure but think that including the Pettitt scores here adds too much clutter. I suggest removing that (the right-hand axis, red time series, and threshold lines) and only including the vertical red lines where a change was detected. I think this will make the plots more readable, and the most relevant point about the change detection is if and when it happens and not the precise Petttitt rank score.**

Thanks for your comments and suggestions. We agreed with you and we have revised this figure as shown in follow:

[Figure]

**Figure 7**. Boxplots of annual AOD variations and the change points calculated by TFPW-MK and Pettitt's methods, based on MODIS DT&DB Products. Trends (corrected Sen's slope) in different regions are shown in the top left corner of subplots, and their significances are indicated by asterisks: * $p < 0.05$. Red vertical bars represent the years of change points which are detected by the Pettitt's methods and pass the significance test ($p < 0.05$). It is noted that the change points in SEA, WA, CA, and NA appear around 2016, 2005, 2015, and 2008, but they do not pass the significance test and thus are not shown.

12. **A more general comment: this paper frames a lot of analyses in terms of statistical significance at the 5% level. This is quite a binary way of thinking that is common in science and is something of an arbitrary practice that has become entrenched. One issue which is not really discussed here but is common in the analysis here (and many other papers) is that the study is not doing a single hypothesis test: there are thousands of them in the paper (every single grid point where a significance test is made is a hypothesis test). So it is fine to take the p=0.05 threshold and say we will mask the areas where the odds of seeing a result at least this big if there is no underlying relation is 5% or less. But the flip side is the false discovery rate – that some of the apparent significant results will belong to those 5%. And due to the high**

**spatiotemporal autocorrelation in aerosol data, some of these will be clustered and so look realistic as opposed to noise or artefacts. The danger is then in overinterpreting something which looks realistic but may be spurious. One of my favourite papers on this topic is Wilks (2016): https://journals.ametsoc.org/view/journals/bams/97/12/bams-d-15-00267.1.xml That and studies cited within suggest methods to control the false discovery rate by adjusting the p-value threshold chosen, so you can be more sure that apparent significant results are real. Even if that is not done, though, I think it is important that this aspect is acknowledged and discussed in the paper as many readers might be statistically unaware. So in general I prefer to focus less on p-values and more on uncertainty estimates for e.g. trends – an "insignificant" trend could be because the likely trend is small and its uncertainty is also fairly small, or because the trend uncertainty is large (in which case we might not have a good estimate of its magnitude or sign). These two are quite different situations. For example in this study the MERRA2 trend maps have a lot more data labelled significant than MODIS – this is expected because MERRA2 is spatiotemporally complete which means the uncertainty of the trend is lower so confidence in the estimate is generally higher (so there is more statistical power to detect a trend and you can see smaller ones).**

We really appreciate your comments and suggestions here. In the previous version of the manuscript, we focused on the problems caused by temporal autocorrelation but ignored spatial autocorrelation problem and the false discovery rate. After studying related publications such as Wilks (2016) [1] that you pointed out, we realize that this is an important technology to help us reduce possible fake trends, and getting a rigorous and cautious trend result is exactly what we want to do. These trend analysis and related constraint methods considering spatiotemporal autocorrelation are worth popularizing. Therefore, following your suggestions, we have added the false discovery rate test [2] to our study and a short paragraph to explain the importance of it. From the results, the adding of false discovery rate test do not affect our main conclusions, but it does increase readability and reduce some false trends that look like fragments.

For trend analysis in regional scale (such as EA, SEA, SA, et al.), we do not apply the false discovery rate test because the spatial autocorrelation is probably negligible in these cases. Instead, we added multiple significance standards (p < 0.01, p < 0.05, or p < 0.10) to enrich study content.

The specific revisions are shown below:

(Lines 100-104) And, the spatial auto-correlation is also a critical problem (Wilks 2016), especially in trend mapping based on a grid point. Because every single grid point will not be completely independent in the significance test. Some of these will be clustered and look realistic as opposed to noise, due to potential spatiotemporal auto-correlation in aerosol data. In this case, the evaluation of the collective significance of a finite set of individual significance tests is recommended (Livezey and Chen 1983), which is usually ignored in previous studies.

(Lines 266-278) Generally, in a multiple hypothesis testing, the noise (true null hypothesis is erroneously rejected) will be unavoidable and the amount depends on selected different threshold of significant level. Especially in geophysical studies, because the spatial auto-correlation between the geophysical data is common (Wilks 2006), the noise will tend to cluster in somewhere to make it seem real and probably lead to a false conclusion. The False Discovery Rate (FDR) test (Benjamini and Hochberg 1995) can provide a good solution to this problem. It allows the use of a dynamic threshold for statistical significance test, with the goal of minimizing the Type I error, as Eq. 11.

$$p_i = \frac{i}{N} \cdot \alpha_{FDR} \tag{11}$$

where, $N$ is the total number of multiple hypothesis testing. $i$ is the position in local p-value rank (from small to large). $\alpha_{FDR}$ is the false discovery rate ($\alpha_{FDR} < 0.05$ in this study), and $p_i$ is the threshold of significance test for each local hypothesis. As the equation shows, when the p-value of a local hypothesis is not small enough to satisfy its position in the whole, it will be removed. Moreover, this FDR test has been proven to be effective and robust when processing the spatial auto-correlation geophysical data, which is also recommended (Wilks 2016). By this way, we can reduce uncertainty, avoid over-interpreting, and more sure that apparent significant trends are real.

[Figure]

**Figure 9.** Long-term trends of seasonal AOD, SAOD, MAOD, and LAOD in Asia from 2000 to 2020. The total AOD is from MODIS C6.1 DT&DB dataset and the size-segregated AOD is from MISR V23 aerosol products. All the points shown in the figure have passed the FDR test ($\alpha_{FDR} < 0.05$). Gray colour represents that the trend is not significant.

[Figure]

**Figure 10.** Long-term trends of seasonal sulphate, dust, SS, OC, and BC AOD in Asia from 2000 to 2020. These type-segregated AODs are obtained from MERRA-2 reanalysis data. All the points have passed the FDR test ($\alpha_{FDR}$ < 0.05). Gray colour represents that the trend is not significant.

[1] Wilks, D.S.: "The Stippling Shows Statistically Significant Grid Points": How Research Results are Routinely Overstated and Overinterpreted, and What to Do about It. Bulletin of the American Meteorological Society, 97, 2263-2273, doi:https://doi.org/10.1175/BAMS-D-15-00267.1, 2016

[2] Livezey, R.E., & Chen, W.Y.: Statistical Field Significance and its Determination by Monte Carlo Techniques. Monthly Weather Review, 111, 46-59, doi:https://doi.org/10.1175/1520-0493(1983)111<0046:SFSAID>2.0.CO;2, 1983

---

## Author Response (AR2)

**Respond to Reviewer and Editor**

Dear reviewer and editor, thank you for your useful comments. We are very willing to revisit these outstanding comments on this version of revision. For the phrasing/grammatical issues, we will also check them with the language editor in the next step before the probably publishing to make sure that all our manuscript is OK.

We have carefully checked the comments, and replied to your comments point by point and included corresponding modifications. In the following text, your comments are marked in bold italics, our responses are in black, and the modifications in manuscript are shown in blue.

*I was a reviewer of the previous version of this manuscript. For the previous version, I liked the study but had some statistical and presentation concerns. In this version the authors have put in a lot of effort revising and have addressed my comments well. As with the previous version of this paper, journal copy-editing will be necessary to correct some phrasing/grammatical issues throughout but the paper is readable and the authors did a good job with this considering they are likely not native English speakers.*
*I have a few minor suggestions based on this revised version, but otherwise am happy for the manuscript to be published after these are taken care of. I would be happy to look at it again if the Editor would like, but I don't know that it would be necessary.*

Dear reviewer, thank you for your recognition and understanding. Your comments are very useful for improving our manuscript.

*1. Line 62: the wording here is awkward. I think the point the authors are trying to make here is that satellites can make observations globally (even if the coverage of products is not quite global). In that case I might change the last part of this sentence to "collect aerosol distribution information from space near-globally with spatial resolutions typically of the order of km".*

Dear reviewer, thank you for your comments. Here, we want to express that comparing to ground-based sites such as AERONET, the observations from satellite are continuous in spatial scale. To avoid possible misunderstanding from readers, we have revised this sentence.

Line 62-64: Satellite observations are an irreplaceable approach because they can collect aerosol distribution information from space near-globally and greatly make up for the limitation of **the cover** in spatial scale, compared with ground-based observations.

*2. Lines 135-136: I am not sure what the sentence about bias and uncertainty refer to here. The AOD uncertainty in the mid-visible from AERONET is about 0.01. I wonder if the authors are talking about the difference between level 1.5 and level 2.0 AERONET direct Sun data? I don't think that is necessary – the rest of the text around there seems to be about the inversion product and not the direct Sun data – and it's best just to give the main level 2.0 direct Sun AOD uncertainty around 0.01.*

Dear reviewer, thank you for your comments. We have revised this part following your suggestions to improve our manuscript.

Line 137: The uncertainty of AOD calculation is reported as ~0.01.
Line 138-143: In addition, an extra site established at Wuhan University was also applied in this study (Jin et al. 2021)

to supplement the aerosol optical properties in CC area. This site is equipped with the same sun sky photometer as AERONET and calibrated annually using the China Meteorological Administration Aerosol Remote Sensing Network (CARSNET) (Che et al. 2009) to ensure the data quality. The AOD from Wuhan site was calculated from the direct sun measurement and other complex properties were retrieved from the sky irradiance under cloudless conditions (Smirnov et al. 2000), with the method of Dubovik and King (2000).

**3. Table 1: I think the rho header column is not aligned correctly here (mentioning in case the typesetters do not catch this).**

Dear reviewer, thanks for pointing out this mistake. We have checked and revised it.

**4. Figure 8: I don't think that three different significance levels need to be highlighted here (\*, \*\*, and \*\*\*). I would just pick one (maybe 0.05 for consistency with elsewhere) or report trend and uncertainty estimate (for all panels) instead so the reader can assess confidence. I would also delete the last sentence in the caption "It is noted..." since if there is no visually-apparent or statistically detectable change point I don't think it needs to be mentioned.**

Dear reviewer, thank you for your comments. I have revised this figure. Yes, as you pointing out, we should emphasize findings which are significant.

**5. Table 2: As in the Figure 8 comment, I'd just stick to one marker of significance and not three. If you're selecting as threshold as significant vs. not significant, I don't think it is meaningful to add extra subcategories, and it makes it harder to read at a glance.**

Dear reviewer, thank you for your comments. I have revised this table. Now, we use a uniform significance criterion that only trends passing the significance test ($p < 0.05$) are shown in it.

**6. Line 632: It is ok to say that the aerosols are fairly stable, but I don't think it directly means it is anthropogenic. It may well be in this case if it's agricultural burning, but the wording implies that stability means anthropogenic, which is not the same thing. I would just say it's stable or maybe that it implies the burning is fairly stable.**

Dear reviewer, thank you for your comments. In this section, we focus more on showing the actual variability of aerosols than on explaining the causes of this variability. Of course, to avoid possible misunderstanding, we carefully checked this section and revised following your comments.

Line 628-629: In contrast, burning in spring and autumn in SEA did not lead to dramatic changes in relative AOD variability. This means that the aerosols in this region are dominated by the long-term and stable burning events.